# Auranofin targets UBA1 and enhances UBA1 activity by facilitating ubiquitin trans-thioesterification to E2 ubiquitin-conjugating enzymes

Wenjing Yan[1,2], Yongwang Zhong[1,2], Xin Hu[3], Tuan Xu [3], Yinghua Zhang[4], Stephen Kales[3], Yanyan Qu[3], Daniel C. Talley[3], Bolormaa Baljinnyam[3], Christopher A. LeClair [3], Anton Simeonov[3], Brian M. Polster [5], Ruili Huang[3], Yihong Ye[6], Ganesha Rai[3], Mark J. Henderson [3], Dingyin Tao [3] ✉ & Shengyun Fang [1,2,7] ✉

UBA1 is the primary E1 ubiquitin-activating enzyme responsible for generation of activated ubiquitin required for ubiquitination, a process that regulates stability and function of numerous proteins. Decreased or insufficient ubiquitination can cause or drive aging and many diseases. Therefore, a small-molecule enhancing UBA1 activity could have broad therapeutic potential. Here we report that auranofin, a drug approved for the treatment of rheumatoid arthritis, is a potent UBA1 activity enhancer. Auranofin binds to the UBA1's ubiquitin fold domain and conjugates to Cys1039 residue. The binding enhances UBA1 interactions with at least 20 different E2 ubiquitin-conjugating enzymes, facilitating ubiquitin charging to E2 and increasing the activities of seven representative E3s in vitro. Auranofin promotes ubiquitination and degradation of misfolded ER proteins during ER-associated degradation in cells at low nanomolar concentrations. It also facilitates outer mitochondrial membrane-associated degradation. These findings suggest that auranofin can serve as a much-needed tool for UBA1 research and therapeutic exploration.

A cascade of three enzymes, comprising E1 ubiquitin-activating enzyme, E2 ubiquitin-conjugating enzyme, and E3 ubiquitin ligase, catalyzes protein ubiquitination. UBA1 is the primary E1 for ubiquitin activation[1,2]. Another E1 enzyme, UBA6 is a dual-activity enzyme that activates both ubiquitin and the ubiquitin-like protein FAT10[3]. E1 activates ubiquitin through C-terminal adenylation, thioester bond

formation with E1 catalytic cysteine, and thioester bond transfer (trans-thioesterification) to E2 that is associated with the ubiquitin fold domain (UFD) of E1[4]. Ubiquitin-loaded E2 is subsequently released from E1 and binds to an E3 ubiquitin ligase, which recruits a substrate protein and catalyzes the transfer of ubiquitin from E2 via a HECT domain or directly to the substrate protein by a RING finger[5,6]. UBA1

[1]Center for Biomedical Engineering and Technology, University of Maryland School of Medicine, Baltimore, MD 21201, USA. [2]Department of Physiology, University of Maryland School of Medicine, Baltimore, MD 21201, USA. [3]National Center for Advancing Translational Sciences, National Institutes of Health, Rockville, MD 20850, USA. [4]Center for Innovative Biomedical Resources, Biosensor Core, University of Maryland School of Medicine, Baltimore, MD 21201, USA. [5]Department of Anesthesiology and Center for Shock, Trauma and Anesthesiology Research (STAR), University of Maryland School of Medicine, Baltimore, MD 21201, USA. [6]Laboratory of Molecular Biology, National Institute of Diabetes and Digestive and Kidney Diseases, National Institutes of Health, Bethesda, MD 20892, USA. [7]Program in Oncology, UM Greenebaum Comprehensive Cancer Center, University of Maryland School of Medicine, Baltimore, MD 21201, USA. ✉e-mail: dingyin.tao@nih.gov; sfang@som.umaryland.edu

cooperates with 30 different E2s and about 600 distinct E3s to catalyze the ubiquitination of hundreds of substrate proteins[1,7–10], which regulates a wide range of cellular functions, such as signal transduction, gene transcription, DNA repair, cell cycle progression, apoptosis, protein quality control, and protein trafficking.

Impaired UBA1 activity or decreased ubiquitination causes or drives several severe human diseases. For instance, the autosomal recessive neuromuscular disease spinal muscular atrophy (SMA) is caused by loss of survival motor neuron 1 (SMN1) protein accompanied by a downregulation of UBA1 expression[11]. Systemic restoration of UBA1 by AAV9-UBA1-mediated expression in SMA mice increased survival and motor performance, and improved neuromuscular and organ pathology, suggesting that loss of UBA1 plays a key role in SMA pathogenesis[11]. Germline mutations of *UBA1* that reduce its enzymatic activity causes the X-linked infantile spinal muscular atrophy (XL-SMA)[12–14]. Children with XL-SMA usually do not survive past early childhood due to respiratory failure. Somatic mutations of *UBA1* in hematopoietic stem cells have been identified as the causative factor for VEXAS syndrome, a severe adult-onset inflammatory syndrome that is often fatal[15]. The majority of VEXAS-causing *UBA1* mutations are found in the codon for methionine 41, which results in loss of the canonical cytoplasmic isoform of UBA1 (UBA1b) and leads to expression of a catalytically impaired isoform initiated at methionine 67 (UBA1c)[15]. Somatic mutations causing VEXAS syndrome have also been identified throughout the UBA1 gene and some of these mutations may be present in germline[16–18]. In addition, *UBA1* mutations have been found to be a driver for a subgroup of lung cancer in never smokers (LINS) but the effect of the mutations on UBA1 catalytic activity has not been examined[19]. Genetic alterations of E3 ubiquitin ligases with impaired E3 activities also underlie the development of several severe diseases. For example, genetic mutations of the *UBE3A* gene encoding the E3 ubiquitin ligase E6AP causes Angelman syndrome[20,21]. Interestingly, small-molecule activators of Angelman syndrome-causing E6AP mutants have recently been reported[22]. Decreased expression of

RNF20 and RNF40 E3 complex has been associated with the development of inflammatory bowel disease and ovarian cancer[23–25].

A major function of ubiquitination is to direct soluble unwanted, damaged, and misfolded proteins to the proteasome for degradation, while sending aggregate-prone proteins to the autophagosome-lysosomal pathway for removal[26]. Decreases in protein clearance play a key role in the development or progression of aging and age-related neurodegenerative diseases, such as Alzheimer's, Parkinson's and Huntington's disease as well as amyotrophic lateral sclerosis[27–29]. These associations of impaired UBA1 activity and insufficient ubiquitination with diseases and aging suggest that enhancing UBA1 activity could have broad therapeutic potential. Although UBA1 inhibitors have been developed for years[30–32], no UBA1 activity enhancer has so far been reported.

In this study, we report that auranofin (AF), a drug currently used to treat rheumatoid arthritis, is a potent enhancer of UBA1 activity. AF increases UBA1 activity at concentrations roughly 4.5 to 73 times lower than the maximum serum concentration (*Cmax*, 459.8 nM) achieved by the approved therapeutic dose for rheumatoid arthritis[33]. Thus, this study discovered a much-needed tool for UBA1 research and therapeutic exploration.

## Results

### Identification of UBA1 as a molecule target of AF

In search for small-molecule modulators of ER-associated degradation (ERAD), we found that AF at 100 nM increased endogenous UBA1 and UBE2G2 interaction in a reciprocal co-immunoprecipitation (coIP) study (Fig. 1a–c). Using surface plasmon resonance (SPR), we then investigated if one of these two interacting proteins is a direct target of AF. The analysis revealed that AF binds to immobilized recombinant UBA1 but not to UBE2G2 (Supplementary Fig. 3a, b). As shown in Supplementary Fig. 3a, single-cycle kinetics was performed for UBA1, due to the difficulty in binding surface regeneration, which is consistent with the fact that AF is known to covalently bind to its target

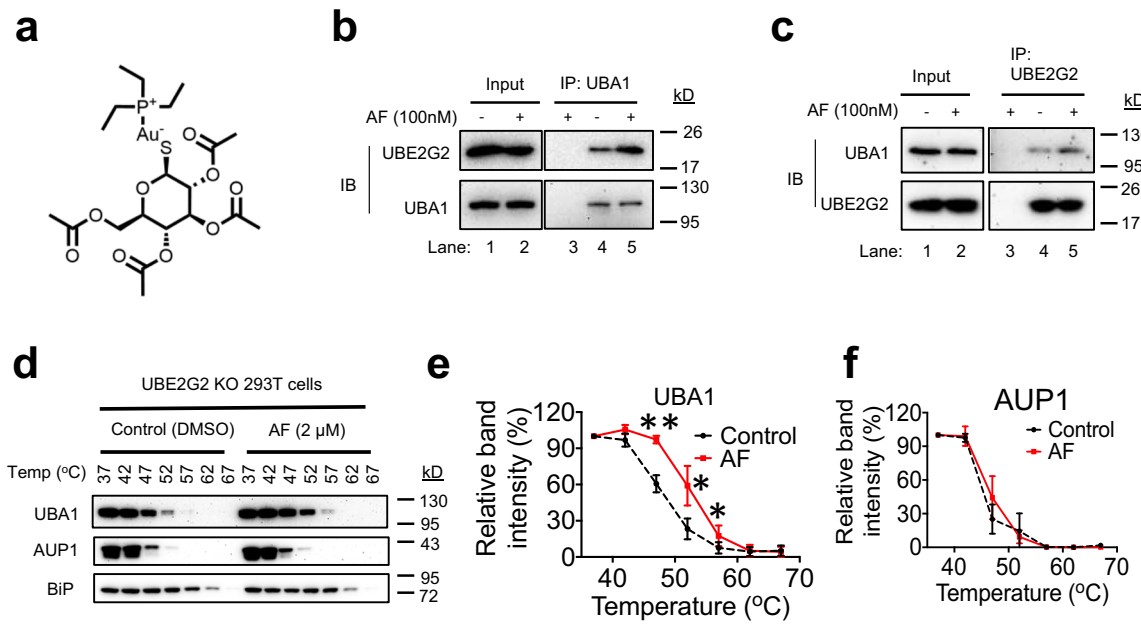

**Fig. 1 | Auranofin (AF) binds to UBA1 and enhances UBA1 interaction with UBE2G2. a** Chemical structure of AF. **b, c** AF enhances UBA1 interaction with UBE2G2 in cells. Total lysates from 293 T cells treated with 100 nM AF for 3 h were used for anti-UBA1 (**a**) or anti-UBE2G2 (**b**) co-immunoprecipitation (coIP) assays. Lane 3 is anti-IgG control. **d** Effects of AF on the thermal stability of UBA1 and AUP1 in UBE2G2 knockout 293 T cells as revealed by CETSA. The indicated proteins were

blotted in soluble fractions. **e, f** The intensities of the bands for UBA1 and AUP1 in (d) were quantified. Data are presented as mean values ± S.D., $n = 3$ biologically independent experiments. *p*-value was calculated by two-tailed paired *t*-test. *$p < 0.05$ and **$p < 0.01$ for UBA1. $p = 0.008$, $p = 0.0421$ and $p = 0.0285$ at 47, 52 and 57 °C, respectively. No difference for AUP1. Source data are provided as a Source Data file.

proteins[34]. The covalent attachment of AF to UBA1 most likely limits the binding surface regeneration. The cellular thermal shift assay (CETSA) was then used to examine whether AF binds to UBA1 in cells[35]. CETSA is based on the biophysical principle of ligand-induced thermal stabilization of its target protein. Ligand-stabilized protein target can be detected in soluble cellular fraction by immunoblotting (IB) or mass spectrometry[35,36]. Since small molecule can stabilize its target and target-binding proteins in CETSA, to eliminate the effects of UBE2G2, CETSA was performed in UBE2G2 knockout (KO) 293 T cells. The treatment with AF-enhanced the thermal stability of UBA1 but not the control protein, ancient ubiquitous protein 1 (AUP1) (Fig. 1d–f), suggesting that AF interacts with UBA1 in cells. These findings imply that UBA1 is a molecular target for AF, and that AF binding improves UBA1-UBE2G2 interaction.

UBA1's C-terminal ubiquitin fold domain (UFD, aa942-1058) interacts with E2s, which is required for ubiquitin trans-thioesterification to E2s during ubiquitination[9,37–40]. Thus, we hypothesized that AF binding to the UFD was responsible for enhancing the UBA1-UBE2G2 interaction. As a positive control, in vitro 6His-UBA1 pull-down assay was performed and demonstrated that AF enhances UBA1-UBE2G2 interaction in a dose-dependent fashion (Supplementary Fig. 1a). In an in vitro maltose-binding protein (MBP) fusion of UBA1's UFD of (MBP-UFD) pull-down assay, AF did enhance the interaction between MBP-UFD and UBE2G2 in a dose-dependent manner (Supplementary Fig. 1b). As a control, MBP fusion of a 220 amino acid UBA1 N-terminal fragment did bind to UBE2G2 and AF also had no effect (Supplementary Fig. 1b). It is worth noting that 6His-UBA1 and MBP-UFD were preincubated with AF and unbound AF was removed before being used in the pull-down assays (Supplementary Fig. 1), consistent with an irreversible effect of AF. AF is known to covalently conjugate to cysteine of previously reported target proteins[34,41–44]. UFD has two cysteine residues (C1039 and C1040), but only one (C1039) has a side chain exposed to the UFD surface, as reported in the UBA1 structure (PDB ID: 6DC6)[45]. To determine whether C1039 is the site for AF conjugation, purified recombinant UBA1 protein treated with AF or DMSO as a negative control were subjected to trypsin digestion followed by HPLC–MS/MS to search for an AF conjugated amino acid. The results showed that only one tryptic peptide (aa1025 to 1054) containing C1039 and C1040 was conjugated with one AF molecule (Supplementary Fig. 2), suggesting that C1039 is the highly possible AF conjugation site. Consistently, SPR revealed that AF binds to UBA1(C1040A) but not UBA1(C1039A) (Supplementary Fig. 3). If AF conjugates to cysteine to exert its effect, cysteine alkylation would eliminate the activity. Indeed, pretreatment of MBP-UFD with the alkylation agent iodoacetamide prevented AF-enhanced UFD-UBE2G2 interaction (Fig. 2a). The reducing agent dithiothreitol (DTT) also inhibited the effect of AF (Fig. 2a). To obtain further support in cells, we transiently expressed HA-tagged wt UBA1, UBA1(C1039A), or UBA1(C1040A) in 293 T cells. Anti-HA coIP revealed that the C1039A mutation, but not the C1040A mutation, prevented AF-enhanced UBA1 and UBE2G2 interaction (Fig. 2b). The result was corroborated in in vitro pull-down assays using MBP-UFD and MBP-UFD with cysteine mutation (Fig. 2c). These results suggest that AF binds to UFD through conjugation to C1039.

To gain insight into the binding mechanism of AF with UBA1, we performed docking studies of AF to UFD and modeled the binding complex of AF-bound UBA1 with UBE2G2. The HPLC–MS/MS data showed that AF bound to UBA1 through the Au-PEt3 moiety by forming an adduct S-Au-PEt3 with the thiol group of Cys1039 (Supplementary Fig. 2). The same reaction mechanism of AF with protein has also been reported[46,47]. Consistent with the experimental data, the model shows that AF prefers to binding to a pocket at the UFD domain of UBA1, which is the protein-protein interface for E2 binding. The metal Au covalently binds to residue C1039, while the PEt3 moiety interacts with residues E1037 and E1049 surrounded in the pocket (Fig. 2d).

Moreover, the helix α1 of UBE2G2, which is the main binding domain to E1 proteins[1,48], is oriented into the AF binding pocket and forms extensive H-bonding and hydrophobic interactions with AF and UBA1. To validate the binding model, we performed site-directed mutagenesis studies on the two residues E1037 and E1049. As expected, mutation of either of E1037A or E1049A in UBA1 abrogated the ability of AF to enhance UBA1-UBE2G2 interaction in cells (Fig. 2e). These findings support that AF binds to C1039 site of the C-terminal UFD domain at the E2 binding interface, possibly acting like a molecular glue to stabilize UBA1-UBE2G2 binding interactions.

## AF enhances UBA1 interaction with the majority of the ubiquitin-conjugating E2s

Human genome encodes thirty E2s catalyzing ubiquitin conjugation, including two enzymatically inactive E2 variants, UBE2V1 and UBE2V2[10]. UFD is the common E2 binding site in UBA1. We, therefore, determined the landscape of which E2 interactions with UBA1 would be enhanced by AF, by immuno-purifying UBA1 followed by label-free quantitative mass spectrometry analysis. To facilitate the coIP, HA-UBA1 was transiently expressed in 293 T cells. Cells were then subjected to treatment with AF or vehicle (DMSO) followed by processing for IP for HA-UBA1 and protein identification by mass spectrometry. In two independent experiments with quadruplicates for each condition, the combined results showed that UBA1 coprecipitated with a total of 24 E2s (Fig. 3a). AF increased UBA1 interaction with 20 of the 24 E2s but had no effect on four of them, including UBE2K, UBE2M (E2 for conjugation of ubiquitin-like protein Nedd8), and the two catalytically inactive E2 variants UBE2V1 and UBE2V2 (Fig. 3a). Multiple sequence alignment was used for comparative sequence analysis[49] and a Guide tree was generated to determine the order in which sequences (or groups of sequences) are aligned to each other[49]. The Guide tree of all 36 human E2s showed that the 20 E2s affected by AF are strictly clustered into two families[49–51] (Fig. 3b). None of the E2s for conjugation of ubiquitin-like proteins except for UBE2M (UBC12) were coprecipitated with UBA1, and AF had no effect on UBE2M binding to UBA1 (Fig. 3a), supporting that it enhances UBA1 interaction specifically with ubiquitin-conjugating E2s. This AF activity was confirmed for examples of four E2s in cells by anti-HA-UBA1 coIP of the E2s (Fig. 3c) and endogenous UBA1 coIP with E2s (Fig. 3d), as well as by MBP-UFD pull-down assays with E2s (Supplementary Fig. 4). As seen in UBA1 coIP results (Fig. 3a), AF did not affect UBE2K binding to MBP-UFD (Supplementary Fig. 4). C1039A mutation also abrogated the activity of AF in enhancing UBA1 interaction with UBE2A and UBE2L3 (Fig. 3e). To determine the specificity of AF's effect on UBA1, the effects of AF on the interactions between UBA6, another ubiquitin-activating E1, and its cognate E2 UBE2Z and between UBA2 and UBE2I (UBC9), the E1-E2 pair for catalytic SUMOylation, were investigated in a GST-UBA6 pull-down assay and anti-HA-UBE2I coIP, respectively (Fig. 3f, g). AF had no effect on these interactions (Fig. 3f, g). These results suggest that AF selectively enhances UBA1 interactions with at least 20 ubiquitin-specific E2s, which are clustered into two distinct families.

## AF facilitates ubiquitin trans-thioesterification (charging) to E2s

The function of UBA1 is to activate ubiquitin and transfer the activated ubiquitin to its active cysteine (ubiquitin charging to UBA1) and then to further transfer ubiquitin to the active cysteine of E2 through trans-thioesterification (ubiquitin charging to E2). E2 binding to UBA1 is required for ubiquitin charging. We, therefore, determined whether enhancing UBA1 and E2 interaction by AF affects ubiquitin charging to UBA1 and/or E2 using UBE2G2 as an example in in vitro ubiquitin charging assays. AF

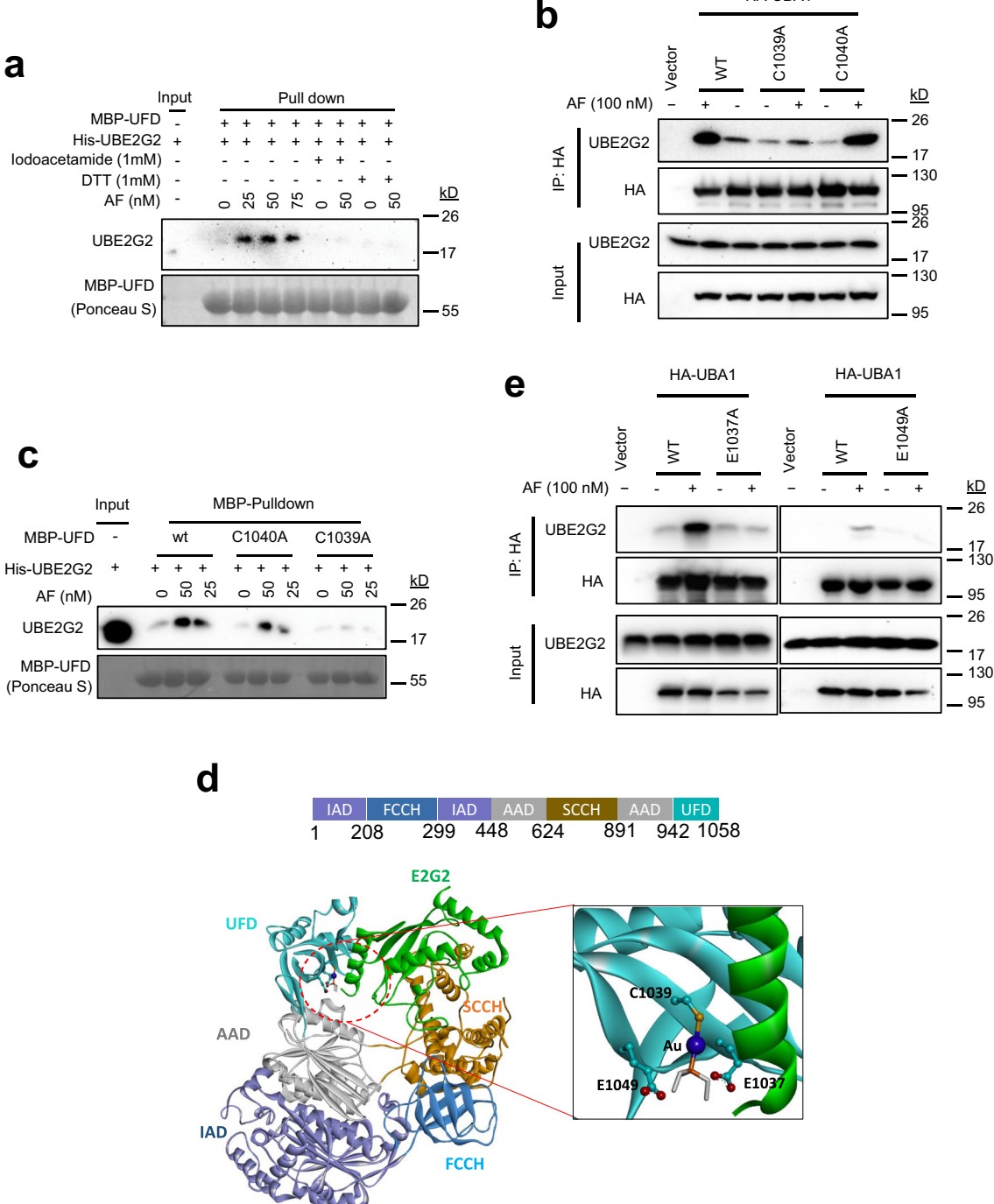

**Fig. 2 | AF binds to the UFD of UBA1 and conjugates to cysteine 1039. a** Effect of Iodoacetamide or DTT on the interaction of UBE2G2 with the UFD fragment of UBA1. **b** C1039A but not C1040A mutation diminishes AF-enhanced UBA1-UBE2G2 interaction in cells. HeLa cells transfected with the indicated plasmids were treated with 100 nM AF for 2 h, followed by processing for IP and IB. **c** C1039A but not C1040A mutation diminishes AF-enhanced MBP-UFD and UBE2G2 interaction in vitro. Recombinant MBP-UFD and its mutants were incubated with UBE2G2 along with different doses of AF. UBE2G2 bound to MBP-UFD, or its mutants was detected by IB. **d** Predicted binding complex of UBA1 with auranofin and E2G2. The binding interaction of the portion of Au-PEt3 of auranofin at the UFD domain is shown in a close-up panel. The protein structure of UBA1 (PDB 6DC6) and E2G2 (PDB 4LAD) are represented in ribbons. The gold Au (I) is coordinated to Cys1039 and rendered as balls in blue, the triethylphosphine ligand of auranofin in the pocket is shown in sticks. The domain structure of UBA1 is shown on top of the model. **e** Mutation of two predicted AF binding residues, E1037 and E1049, respectively, abrogates AF-enhanced UBA1-UBE2G2 interaction by coIP. Source data are provided as a Source Data file.

did not affect ubiquitin charging to UBA1 (Fig. 4a) but clearly increased ubiquitin charging to UBE2G2 in a dose and time-dependent manner (Fig. 4b and S5). Ubiquitin charged on UBE2G2 could be removed by the reducing agent DTT (Fig. 4c), indicating that the ubiquitin is indeed linked to UBE2G2 via a thioester bond. AF also increased ubiquitin charging to a second example of E2s, UBE2D1 (Fig. 4d), suggesting that AF generally enhances UBA1-E2 interaction, resulting in increased ubiquitin charging to E2s.

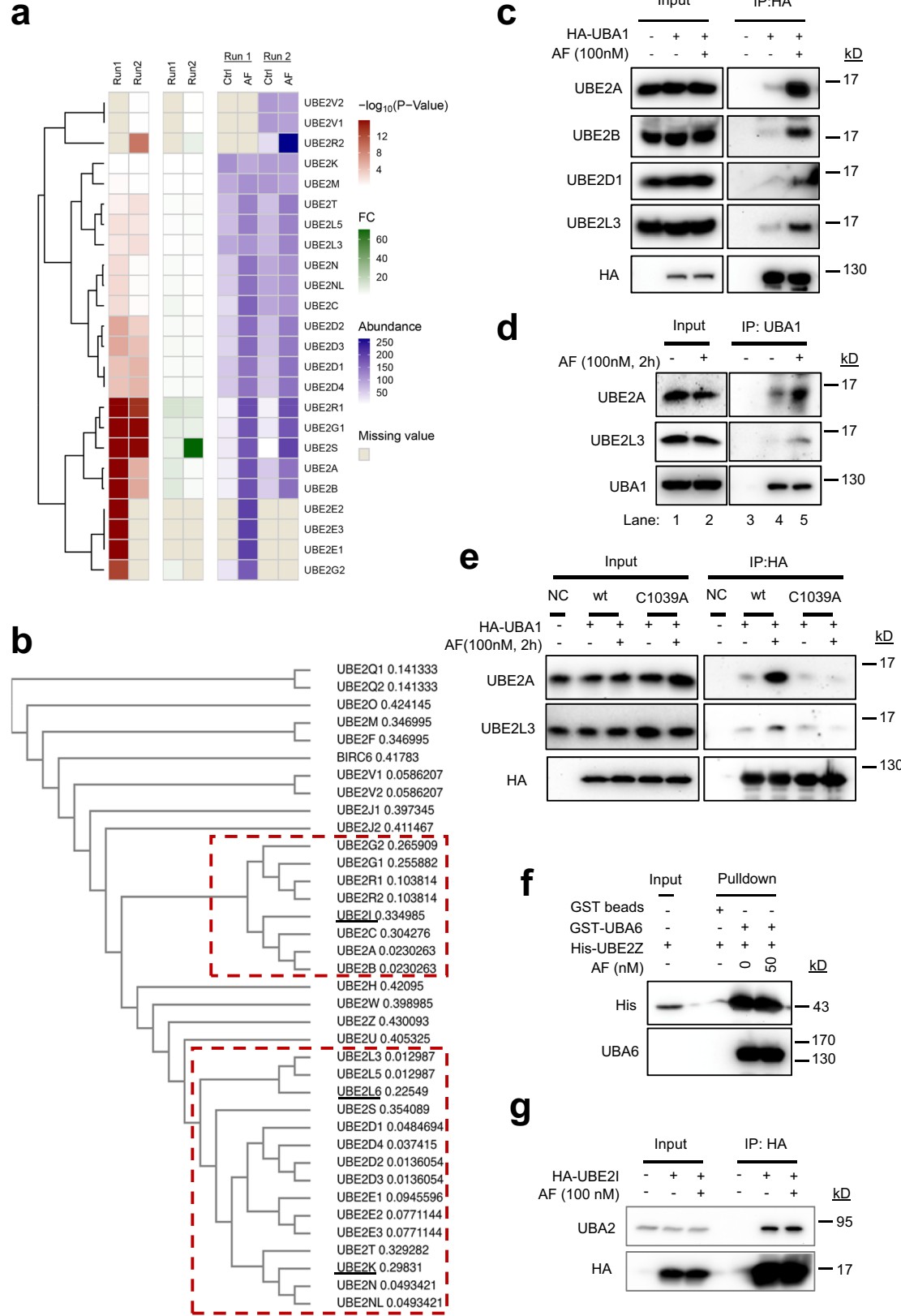

## AF promotes E3 activities and protein ubiquitination and degradation

Ubiquitin charging to E2 is a key step in the ubiquitination enzymatic cascade. Therefore, the effects of AF on the activities of a set of seven representative E3s were investigated in vitro, including the RING finger E3s, gp78, Hrd1, RNF2, and RNF126, the HECT domain E3, E6AP, the

Elongin B/Elongin C/VHL/CUL2/RBX1 (RBX) E3 complex, and the RING-Between-RING (RBR) E3 parkin (Fig. 5a and Supplementary Fig. 6b, c). These E3s regulate diverse cellular functions. gp78 and Hrd1 are the two major E3s for ubiquitination and degradation of misfolded ER proteins during ERAD[52–55]. RNF2 is the main E3 for ubiquitination of histone H2A[56,57]. RNF126 is a quality control E3 for ER proteins

**Fig. 3 | AF enhances UBA1 interaction with most of the ubiquitin-conjugating E2s. a** AF enhances UBA1 interactions with 20 E2s in cells. The heatmap was obtained by two independent ani-HA-UBA1 coIP followed by protein identification and quantification by mass spectrometry. The first four columns represent the summary *p*-value (columns 1,2) and fold change (columns 3,4) for control versus AF treatment groups. Columns 4–8 represent the UBA1-associated protein levels, calculated as an average from quadruplicate samples in each experimental run. **b** Guide tree of 36 human E2 ubiquitin-conjugating enzymes. Multiple sequence alignment and Guide tree of the 36 E2 proteins was performed by Clustal Omega[39]. The E2s in two clusters in red dash-line rectangles are E2s whose interactions with

UBA1 were enhanced by AF as shown in (**a**). The three E2s underlined are exceptions. The number following the E2s is indicative of the evolutionary distance between the sequences. **c** Validation of AF-enhanced UBA1-E2 interactions in cells by anti-HA-UBA1 coIP as in (**a**) followed by IB. **d** Validation of AF-enhanced UBA1-E2 interactions in cells by anti-UBA1 coIP as in (**a**) followed by IB. **e** C1039A mutation diminishes UBA1 interactions with E2s. **f** AF does not promote the UBA6 interaction with UBE2Z in a GST-UBA6 pull-down assay. **g** AF does not affect the UBA2 interaction with UBE2I as revealed by anti-HA-UBE2I coIP. Source data are provided as a Source Data file.

mislocalized to the cytosol and assists ubiquitination of dislocated ER proteins during ERAD[58,59]. E6AP ubiquitinates multiple substrate proteins[60]. Notably, human papillomavirus (HPV) hijacks E6AP to ubiquitinate the tumor suppressor protein p53, adding viral infection and contributing to cervical cancer[60,61]. The RBX E3 complex ubiquitinates inhibitor of kappa light polypeptide gene enhancer in B-cells, kinase beta (IKBKB) and NRF2[62,63]. Parkin plays a key role in mitophagy[64]. AF potently enhanced the activities of all seven E3s as measured by in vitro E3 activity assay using either full-length GST-fusion proteins or their E3-active domain-containing fragments (Fig. 5a and Supplementary Fig. 6a–d). As predicted, AF failed to enhance UBA6 activity in ubiquitin charging to UBE2Z (Supplementary Fig. 6e). Interestingly, AF potently enhanced gp78-catalyzed ubiquitination in the presence of the cytosolic form of UBA1, UBA1b, or the VEXAS syndrome-causing UBA1c, a truncated form of UBA1b that uses methionine 67 as translation start codon due to mutations of codon methionine 41[15] (Fig. 5b, c). These results suggest that AF-enhanced UBA1-E2 interaction and ubiquitin trans-thioesterification to E2 increase in E3 activities.

**AF facilitates ERAD and outer mitochondrial membrane-associated degradation (OMMAD).** One of the major functions of ubiquitination is to target unwanted and misfolded proteins to the proteasome for degradation. In ERAD, ubiquitination is a prerequisite for substrate dislocation by p97/VCP[65,66]. Consistently, we demonstrated that treatment of the UBA1 activity inhibitor TAK-243 abrogated dislocation of two representative ERAD substrates NHK and CD3δ in a dose-dependent manner (Supplementary Fig. 7). As an enhancer of UBA1 activity, we predicted that AF would facilitate the degradation of proteasomal substrates. To test this possibility, we examined the effects of AF on degradation of the well-characterized ER luminal ERAD substrate, the null Hong Kong variant of α−1-antitrypsin (NHK)[55,67,68]. AF at concentrations tested from 25 to 100 nM markedly decreased the levels of NHK in HeLa cells in a dose-dependent manner (Fig. 5d). The decrease was blocked by the proteasome inhibitor bortezomib (BTZ) (Fig. 5e). In addition, AF at 100 nM also increased NHK ubiquitination (Fig. 5f). One of the roles of ubiquitination is to facilitate dislocation of misfolded ER proteins by p97/VCP for delivery to proteasomes for degradation. Using drGFP to report NHK dislocation in live cells, AF exhibited a dose-dependent enhancement of NHK dislocation (Fig. 5g). AF also downregulated CD3δ, an ER-membrane spanning ERAD substrate[67,69] (Fig. 5h). As previously reported, AF at concentrations that enhances ERAD also induce oxidative stress due to inhibition of TrxRs as demonstrated by OxyBlot, which detects carbonyl groups introduced into proteins by oxidative reactions (Supplementary Fig. 8a). However, independently knocking down the known targets for AF, TrxR1 and TrxR2, did not alter the extent of AF -induced NHK degradation or mimic the ability of AF to promote NHK removal (Supplementary Fig. 8b–e), indicating that AF-induced NHK degradation is not mediated by targeting TrxR1 and TrxR2.

To obtain additional support for the activity of AF in promoting protein ubiquitination and degradation, we determined the effects of AF on ubiquitination and degradation of MiD49 and Mcl1, the established substrates for OMMAD[70,71]. As demonstrated by cycloheximide (CHX) chase, AF accelerated the degradation of both proteins (Fig. 6a, b).

The proteasome inhibitor BTZ impeded their degradation enhanced by AF (Fig. 6a, b, lane 11). Myc-MiD49 and myc-Mcl1 were transiently expressed in HeLa cells and anti-myc reIP was then performed to examine whether AF could enhance MiD49 and Mcl1 ubiquitination. AF promoted ubiquitination of both myc-Mcl1 and myc-MiD49 proteins when proteasomal degradation was blocked by BTZ (Fig. 6c, d). These results suggest that AF promotes protein ubiquitination and degradation in cells, which is consistent with its role as an UBA1 activity enhancer.

To determine whether AF enhances ERAD through binding to C1039 in UBA1, homozygous C1039A mutation in UBA1 gene was created in HCT116 cells using CRISPR/Cas9 technology. As predicted, AF failed to downregulate NHK in UBA1(C1039A) cells (Fig. 7a). AF also failed to promote gp78-mediated ubiquitination in vitro when using a recombinant UBA1 protein containing the same C1039A substitution (Fig. 7b). Moreover, AF lost its activity to enhance UBA1 interaction with E2s and NHK ubiquitination and degradation in UBA1(C1039A) cells (Fig. 7c, d). These results provide further support to a model in which AF binds to C1039 and functions as an UBA1 activity enhancer in cells.

## Discussion

Most cellular ubiquitination processes require UBA1 to activate ubiquitin. Abnormal UBA1 activity or ubiquitination causes or drives many human diseases, such as cancer, major neurodegenerative diseases, Angelman syndrome, VEXAS syndrome, and spinal muscular atrophy, as well as aging, highlighting the importance of the discovery of small-molecule modulators of UBA1 activity for research and therapeutic purposes. Although several UBA1 inhibitors, such as PY-41, TAK-243, and PYZD-4409, have been developed[30–32], pharmacologically enhancing UBA1 activity has thus far not been realized. In this study, we discovered that AF, a clinical drug used for the treatment of rheumatoid arthritis, potently enhances UBA1 activity. It acts by binding to the UBA1's UFD and increases UBA1 interaction with E2s, resulting in facilitated ubiquitin trans-thioesterification (charging) to E2s, which markedly boosts the activities of examples of RING finger, HECT domain, RBX, and RBR E3s in vitro. AF also enhances ERAD and OMMAD by facilitating substrate ubiquitination, an effect that was not mediated by its established targets, the TrxRs. Moreover, mutation of the AF binding site C1039 in HCT116 cells renders AF ineffective in enhancing ERAD. The discovery of AF as an UBA1 activity enhancer provides a much-needed tool for UBA1 research and therapeutic exploration. Furthermore, our findings indicate that the UBA1-E2 interaction is a feasible target for developing more specific small-molecule UBA1 activity enhancers.

AF enhances UBA1-E2 binding in cell-based and biochemical assays at concentrations ranging from 5–100 nM, where it facilitates ubiquitin charging to E2s, promotes E3 activity, and accelerates ERAD and OMMAD. Importantly, AF concentrations necessary for this UBA1-directed activity are roughly 4.5 to 73 times lower than the maximum serum concentration (*Cmax*, 459.8 nM) achieved by the approved therapeutic dose for rheumatoid arthritis[33]. In cells, AF selectively increases UBA1 binding to at least 20 different ubiquitination E2s. The interaction between SUMOylation E1 and E2, UBA2

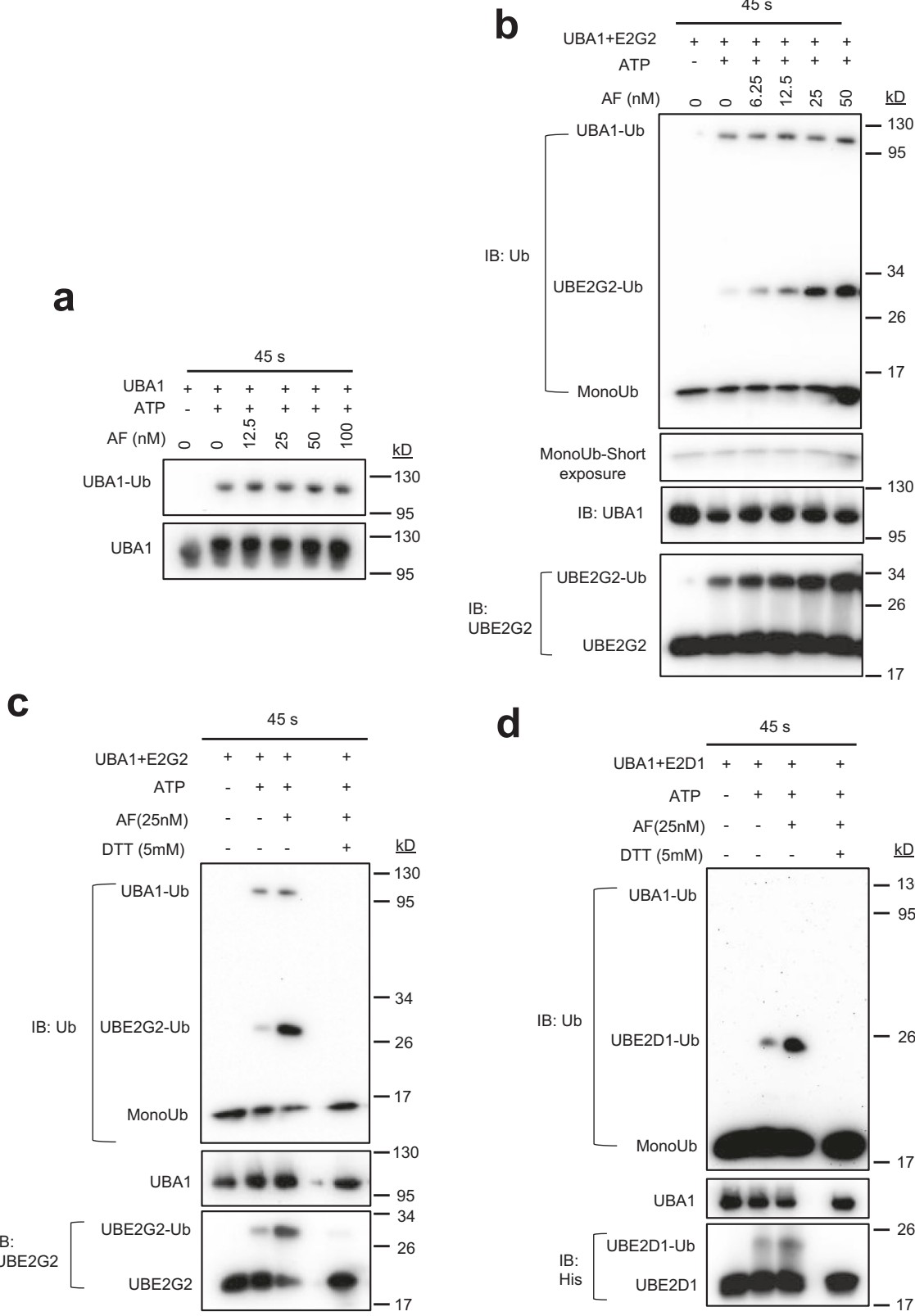

**Fig. 4 | AF facilitates ubiquitin charging to E2s.** The effects of AF on ubiquitin charging in vitro to UBA1 (**a**), UBE2G2 (**b, c**), and UBE2D1 (**d**). The assays were performed using purified recombinant proteins. 250 nM UBA1 alone or in combination with 4 μM His-E2s (His-UBE2G2 or His-UBE2D1) were treated with increasing concentrations of AF in reaction buffer for 1 min. ATP (50 μM) was added to initiate the reaction. The reactions proceeded at 15 °C for 45 s and stopped by adding non-reducing loading buffer. 5 mM DTT was used to disrupt the thioester bond that links ubiquitin to the active cysteine of E2s. Source data are provided as a Source Data file.

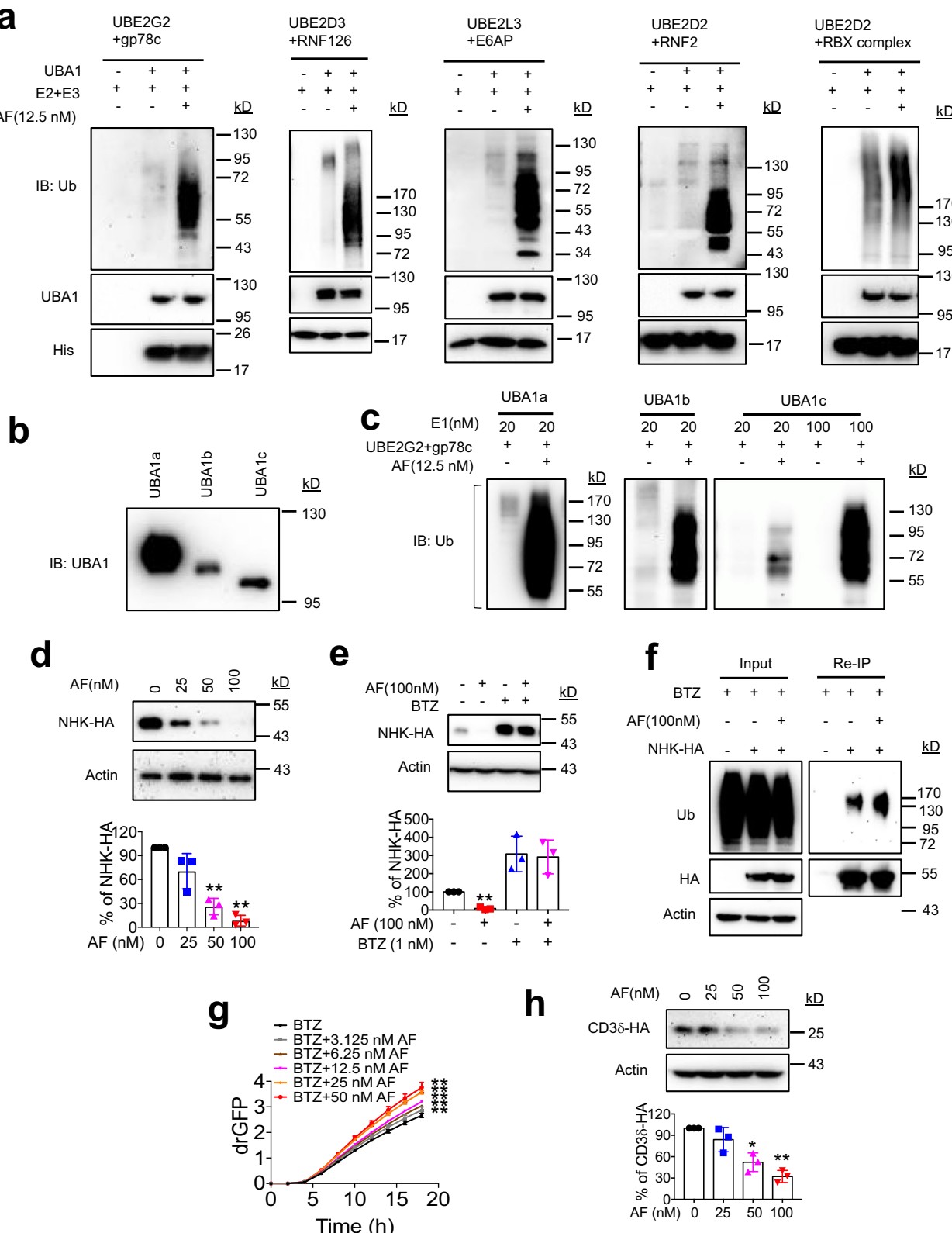

and UBE2I (UBC9), was unaltered by AF, as the interaction between another ubiquitin-activating E1 UBA6 and its cognate E2 UBE2Z. The Guide tree[49–51] of all 36 human E2s indicated that the 20 E2s affected by AF are strictly clustered into two families, implying that these E2s have conserved structural properties that govern their binding to UBA1 can be improved by AF.

The exact mechanism by which AF improves UBA1-E2 binding is unclear. It is known that the UFD of UBA1 interacts with the helix α1 of the core catalytic domain of E2s[72]. Interestingly, we have gathered compelling evidence suggesting the conjugation of AF to C1039 in the UFD. This conjugation on either C1039 or C1040 has been directly demonstrated through mass spectrometry analysis. Notably, the

**Fig. 5 | AF potently enhances E3 activity in vitro and accelerates ERAD in cells.** **a** Effects of AF on E3 activity in in vitro E3 autoubiquitination assay. RBX complex: Elongin B/Elongin C/VHL/CUL2/RBX1. **b** IB of UBA1a (UBA1), UBA1b, and UBA1c used in the E3 activity assay in (**c**). UBA1a, UBA1b, and UBA1c translation start at methionine 1, 41, and 67, respectively. **c** In vitro gp78c autoubiquitination assay in the presence of UBA1a, UBA1b, and UBA1c, respectively, and the effects of AF. **d** AF decreased NHK protein levels in a dose-dependent manner. HeLa cells stably expressing NHK-HA were treated with AF for 24 h and then subject to IB. AF versus Control, $p = 0.1482$, $p = 0.0065$ and $p = 0.0018$ at 25, 50 and 100 μM, respectively. **e** AF-induced NHK downregulation is blocked by proteasome inhibitor, bortezomib (BTZ). AF versus Control, $p = 0.0018$. **f** AF increases NHK-HA ubiquitination in HeLa cells. Anti-HA reIP was performed to determine NHK-HA ubiquitination. **g** AF increases NHK dislocation in a dose-dependent manner. drGFP intensity was monitored and quantified in IncuCyte S3 Live-Cell Analysis System and expressed as mean ± S.D., $n = 4$ wells/treatment. **\*\***$p < 0.01$. AF versus Control, $p = 0.0076$, $p = 0.0086$, $p = 0.008$, $p = 0.007$ and $p = 0.0079$ at 3.125, 6.25, 12.5, 25 and 50 μM, respectively. $p$-values were calculated by two-tailed paired $t$-test. **h** AF decreased CD3δ protein levels in a dose-dependent manner. AF versus Control, $p = 0.2442$, $p = 0.0238$ and $p = 0.0052$ at 25, 50 and 100 μM, respectively The graphs in (d, e, and h) show % increase for each condition relative to DMSO-treated control. Data are presented as mean values ± S.D., $n = 3$ biologically independent experiments. *$p < 0.05$, **$p < 0.01$. $p$-values were calculated by two-tailed paired $t$-test. Source data are provided as a Source Data file.

C1039A but not C1040A mutation abolishes AF's activity in enhancing UBA1-E2 binding and E3 activity in vitro, as measured by SPR, pull-down, and coIP studies. Furthermore, CRISPR/Cas9 substitution of C1039A of the endogenous UBA1 gene abrogated AF's ability to enhance UBA1 activity in cells. C1039 is found at the interface between the UFD of UBA1 and the helix α1 of E2. Modeling of AF binding to the UBA1-UBE2G2 complex revealed that it interacts with residues in both UFD and UBE2G2, implying that AF may act as a molecular glue to improve UBA1-E2 interaction. This molecular glue-like activity may increase ubiquitin charging to E2s, which is reminiscent of the mechanism underlying NEDD8 charging to UBC12[48]. Future structural studies, such as NMR, X-ray crystallography and cryoEM, are necessary to accurately elucidate the binding mechanism of AF to the UBA1-E2 complex. These techniques will provide valuable insights into the precise molecular interactions and spatial arrangement of AF within the UBA1-E2 complex as well as the potential conformational changes in UBA1 induced by AF.

Our data indicates that the increased UBA1-E2 interaction induced by AF promotes ubiquitin trans-thioesterification to E2, resulting in a significant increase in E3 activities. Examining two ubiquitination-dependent protein quality control mechanisms, ERAD and OMMAD, provided evidence that AF also increases UBA1 activity in cells. AF promoted ubiquitination and proteasomal degradation of known model substrates for these two pathways. Although AF at low nano-molar doses also inhibits TrxR1 and 2, knocking down either of these enzymes independently did not mimic or prevent AF-enhanced ERAD. One caveat is that the inhibition of deubiquitinating enzymes (DUBs) can potentially increase proteasomal degradation. A previous study has reported that AF indeed inhibits proteasome-associated DUBs, UCHL5 and USP14. However, this inhibition leads to a suppression of proteasomal degradation[73], which contradicts our observations. Furthermore, a recent study utilized a combination of Thermal-range Thermal Proteome Profiling, Functional Identification of Target by Expression Proteomics, and multiplexed redox proteomics to deconvolute AF targets, and no DUB was identified[74]. Interestingly, their Thermal-range Thermal Proteome Profiling demonstrated an AF-induced thermal shift of UBA1 similar to what we have observed in CETSA; however, no confirmation experiment was conducted in the study to validate UBA1 as AF target[74].

Together, our findings suggest that AF binds directly to UBA1, leading to an enhancement of UBA1 and E2 interaction. The enhanced interaction augments ubiquitin trans-thioesterification to E2s. As a result, cells process a greater abundance of E2s charged with ubiquitin, thereby promoting ubiquitination catalyzed by hundreds of E3s. Moreover, previous studies have shown that ubiquitin-charged E2s exhibit a higher affinity for E3s than ubiquitin-free E2s[75]. Thus, this AF activity provides a significant advantage in the assembly of the ubiquitin-E2, E3, and substrate complex. This complex assembly, in turn, further promotes protein ubiquitination and subsequent degradation.

Due to its inhibitory effect towards TrxRs, AF clearly has limitations as a selective UBA1 activity enhancer. Compromising TrxR function would counteract the potential benefits of increasing UBA1 activity, as it causes oxidative stress and may harm proteins, DNA, and lipids. Fortunately, this study indicates that the UBA1-E2 interaction is a feasible target for the development of specific small-molecule UBA1 activity enhancers, which will be facilitated by further research into the mechanism governing AF-enhanced UBA1 and E2 interaction.

## Methods
### Cell culture
HeLa cells (ATCC, CCL-2), HEK293 cells (ATCC, CRL-1573) and HCT116 cells (ATCC, CCL-247) were obtained from ATCC and cultured in complete Dulbecco's modified Eagle's medium (DMEM) supplemented with 10% fetal bovine serum as growth medium. HeLa cells stably expressing SP-S11-NHK-HA and S1-10 were established previously[67]. UBE2G2 KO 293 T cells were established in Dr. Yihong Ye's laboratory (NIDDK/NIH, USA). JM109 and BL21(DE3) were obtained from New England Biolabs.

### Reagents and antibodies
AF was purchased from Enzo Life Sciences (BML-EI206-0100) and dissolved in DMSO to prepare 5 mM stock solution. All other chemicals and lab reagents were purchased from Sigma–Aldrich unless otherwise indicated.

Recombinant proteins including UBA1 (E-305-025), UBA6 (E-307-025), UBE2G2 (E2-680-100), Elongin B/Elongin C/VHL/CUL2/RBX1 (RBX complex, E3-655), UBE2K (NBP2-35096) and UBE2Z (E2-677-100) were purchased from R&D Systems. Recombinant UBA1a (UBA1), UBA1b, and UBA1c were kindly provided by Drs. Achim Warner (NIH) and David B. Beck (New York University)[15]. UBE2D1 (BML-UW9050), UBE2D2 (BML-UW9060), UBE2D3 (BML-UW9070), UBE2L3 (BML-UW9080) and UBE2N (BML-UW9565) were purchased from Enzo Life Sciences.

The sources of the antibodies are as follows: UBA1 (ab181225, clone EPR14204(B), 1:1000) antibody was from Abcam. UBA1 (A301-125A) antibody for Endogenous IP experiment was from Fortis Life Sciences. ubiquitin-HRP (sc-8017 HRP, clone P4D1, 1:200), anti-TrxR1 (sc-28321, clone B-2, 1:1000), anti-TrxR2 (sc-365714, clone B-10, 1:1000) and anti-UBE2J1 (sc-377002, clone B-6, 1:1000) antibodies were purchased from Santa Cruz Biotechnology. Antibody to MiD49 (28718-1-AP, 1:1000), UBE2B (10733-1-AP, 1:1000) and HRP-conjugated Alpha Tubulin Monoclonal antibody (HRP-66031, clone 1E4C11, 1:3000) were purchased from ProteinTech. UBE2G2 (63182 S, clone D8Z4G, 1:500), Mcl1 (4572 S, 1:1000), UBA6 (13386 S, 1:1000), UBA2 (D15C11, 1:1000) antibodies were purchased from Cell Signaling Technology. UBE2N (101018-T32, 1:1000) and UBE2L3 (108490-T32, 1:1000) were purchased from Sino Biological Inc. Anti-HA-peroxidase (3F10) antibody (12013819001, 1:500) was purchased from Roche. AUP1 (A57895, 1:2000), UBE2A (ABS2203, 1:1000), His (SAB1305538, clone 6AT18, 1:1000), Myc (WH0004609M2, clone 1G7, 1:1000) and β-Actin-Peroxidase antibody (A3854, clone AC-15, 1:3000) antibodies were purchased from Millipore Sigma. Anti-UBE2D1 (NBP3-06652, 1:1000) antibody was purchased from Novus biologicals. BiP (610978, 1:1000) antibody was from BD Biosciences.

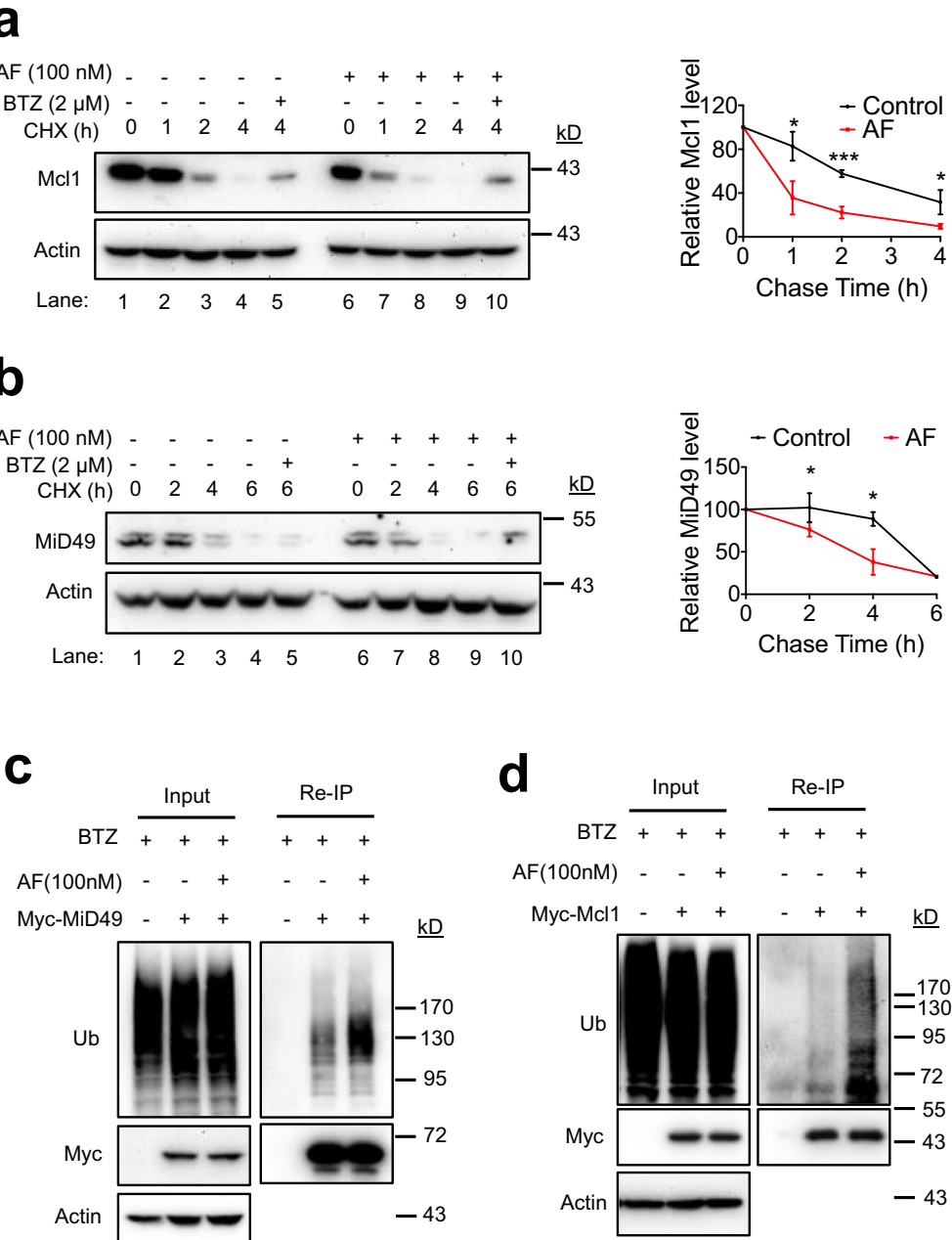

**Fig. 6 | AF accelerates ubiquitination-dependent OMMAD. a, b** Analyses of Mcl1 (**a**) and MiD49 (**b**) degradation using cycloheximide (CHX) chase. HeLa cells were incubated with 100 nM AF and CHX for the indicated durations. The relative band density of Mcl1 or MiD49 was normalized to that of Actin. Data are presented as mean values ± S.D., *n* = 3 biologically independent experiments. \*p < 0.05, \*\*\*p < 0.001. *p*-values were calculated by two-tailed unpaired *t*-test. **c, d** AF promotes MiD49 (**c**) and Mcl1 (**d**) ubiquitination. HeLa cells transient expressing Myc-MiD49 or Myc-Mcl1 were treated with 100 nM AF in the presence or absence of 2 μM BTZ for 4 h. Anti-myc reIP was performed to determine Myc-MiD49 or Myc-Mcl1 ubiquitination. Three biological replicates were performed with one representative experiment shown. Source data are provided as a Source Data file.

### Generation of plasmid constructs for transient transfection

GST-RNF126 (138643) and HA-UBE2I (p3258 pCMV hUBC9 wt HA, 14438) plasmids were purchased from addgene. pGEX-gp78C, pGEX-mE6AP, pGEX-5X-RNF2 and pFlag-CMV-6c-ratUBC7 constructs have been reported[76–78]. Myc-MiD49 and Myc-Mcl1 constructs were kind gifts from Dr. Mariusz Karbowski (University of Maryland, Baltimore)[70,79]. Maltose-binding protein (MBP)-UBA1 (1-220 or 900-1058) plasmids were kindly provided by Dr. Angelos Constantinou (Université de Montpellier, France)[80]. MBP-UFD (942-1058) plasmid was created from MBP-UBA1 (900-1058) using Q5 site-directed mutagenesis kit (New England Biolabs). pCIneo-HA-UBA1 was constructed by inserting the cDNA fragments encoding the ORF of human UBA1

into the NheI/NotI sites of pCIneo. Plasmids encoding mutant HA-UBA1 (C1039A, C1040A, E1037A and E1049A) were generated using a Quik-Change II site-directed mutagenesis kit (Agilent). Related primers are listed in Supplementary Table 1 in Extended data. All expression constructs and mutations were verified by DNA sequencing.

### Purification of recombinant proteins

Constructs encoding GST-tagged gp78C, RNF126, mE6AP, or RNF2 were transformed in *Escherichia coli* BL21/DE3 strains. Single clones of transformed BL21(DE3) transformed were cultured overnight in Lysogeny Broth (LB) medium. Then, the overnight culture was inoculated (1:100) to fresh LB medium and cultured at 37 °C until

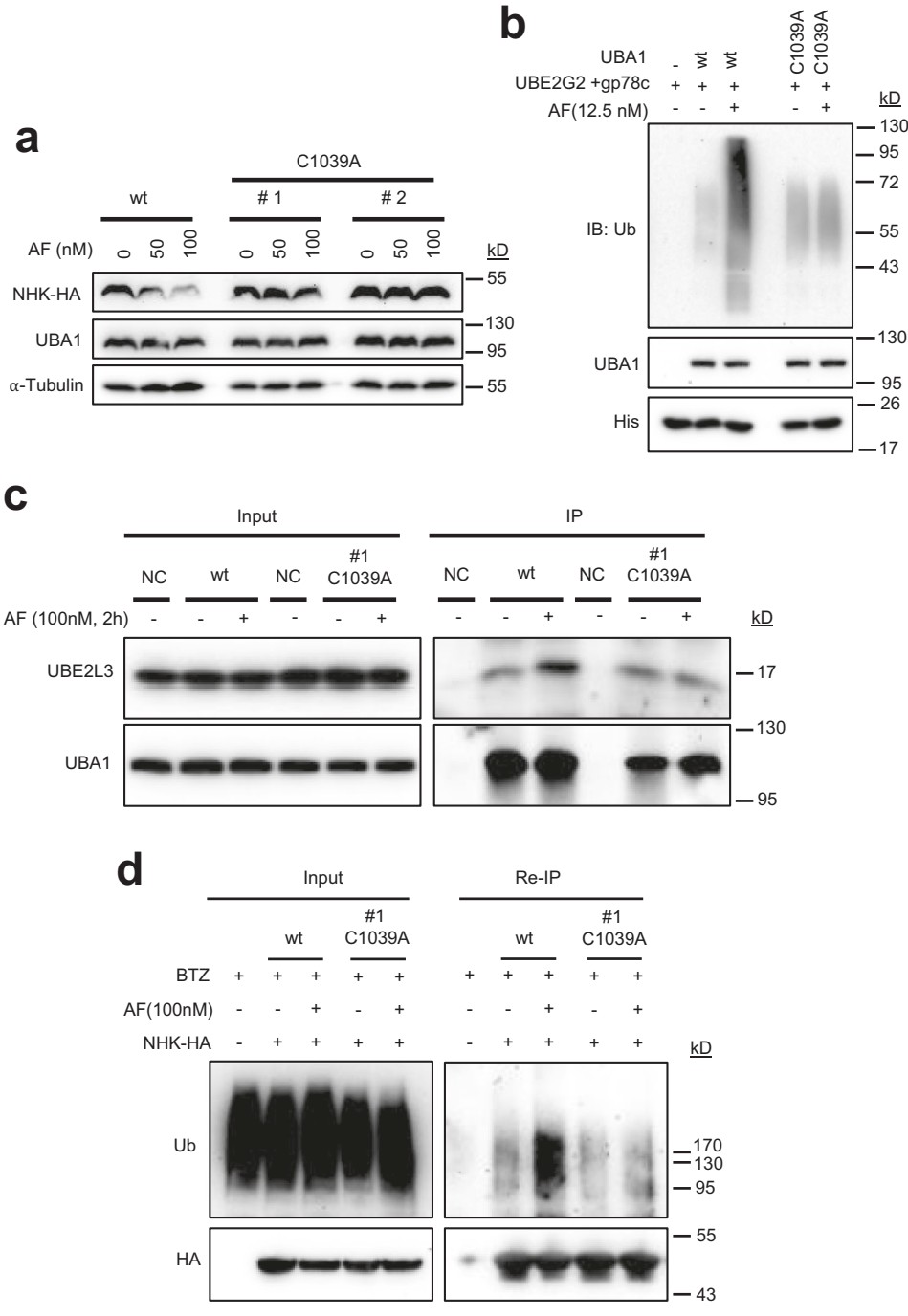

**Fig. 7 | AF does not enhance NHK ubiquitination and proteasomal degradation in HCT116 cells expressing C1039A mutant UBA1 (HCT116-UBA1(C1039A) cells).** **a** AF does not downregulate NHK in HCT116-UBA1(C1039A). HCT116-UBA1(C1039A) cells were generated by CRISPR/cas9. NHK-HA were transiently expressed in parental HCT116 and two different clones of HCT116-UBA1(C1039A) cells. Cells were treated with AF for 24 h and then processed for IB. **b** AF does not enhance gp78c-mediated ubiquitination in vitro in presence of UBA1(C1039A). **c** AF does not increase UBA1-UBE2L3 interaction in HCT116-UBA1(C1039A) cells as revealed by anti-UBA1 coIP. **d** AF does not promote NHK-HA ubiquitination in HCT116-UBA1(C1039A) cells as revealed by anti-HA reIP as described in Fig. 5f. Source data are provided as a Source Data file.

the $OD_{600}$ reached 0.4–0.8. The expression of GST-tagged proteins was induced with 0.2 mM isopropyl β-d-thiogalactoside (IPTG) at 25 °C for 2 h. The bacteria were collected by centrifugation and lysed in lysis buffer (50 mM Tris−HCl, pH 8.0, 1 mM EDTA, 1% Triton X100, 5 mM DTT) containing 200 μg/mL lysozyme with sonication. The lysates were cleared by centrifugation at 20,000 × g for 20 min. To purify the GST-tagged proteins, the cleared lysates were bound to Glutathione Sepharose 4B (GE Healthcare) for 2 h at 4 °C with

rotation. The beads were washed with lysis buffer. Beads containing proteins were used for in vitro ubiquitination assay.

The expression of MBP-UBA1 fragments (1-220 or 942-1058) were induced with 0.2 mM IPTG and 0.2% L-arabinose at 25 °C for 2 h. Bacteria were lysed in lysis buffer (20 mM Tris−HCl, pH 7.4, 200 mM NaCl, 1 mM DTT) containing 200 μg/mL lysozyme with sonication. The supernatant was loaded on amylose resin (New England Biolabs) for 2 h at 4 °C with rotation. The beads were

washed with lysis buffer. Beads containing proteins were used for MBP-pull down assay.

PET21d-UBA1 expression plasmids were transformed into bacterial Rossetta 2(DE3) cells and were grown in LB culture to a cell density of OD600 = 1.2-1.8. Expression was induced by 0.2 mM IPTG induction at 16 °C for 2 h. Cells were pelleted by centrifugation at 4000 × g for 10 min and subsequently resuspended in lysis buffer [50 mM NaH$_2$PO$_4$, 300 mM NaCl, 10 mM imidazole, 10% glycerol, 1% Triton X100, 1 mM DTT, pH 8.0]. Cell lysis was sonicated, and cell debris was removed by ultra-centrifugation at 20,000 × g at 4 °C for 30 min. Clarified lysates containing Uba1 were incubated with pre-equilibrated Ni-NTA agarose resin at 4 °C for 2–4 h. Beads containing UBA1 were washed using wash buffer [50 mM NaH$_2$PO$_4$, 300 mM NaCl, 20 mM imidazole, 10% glycerol, 1% Triton X100, pH 8.0]. Subsequently, UBA1 protein was eluted in wash buffer containing 250 mM Imidazole. UBA1 protein was desalted by dialysis overnight at 4 °C.

### Transient siRNA knockdown and DNA transfection

Negative Control siRNA was purchased from Ambion. All other siRNAs were synthesized by Sigma, including TrxR1 #1 (sense: 5′-GCAAGACUCUCGAAAUUAU[dT][dT]−3′), TrxR1 #2 (sense: 5′-GACAGUUCGUACCAAUUAA[dT][dT]−3′), TrxR1 #3 (sense: 5′-GCGAUAUAUUGGAGGAUAA[dT][dT]−3′), TrxR2 #1 (sense: 5′-GGUUUGCGGCGUUGCCAAA[dT][dT]−3′), TrxR2 #2 (sense: 5′-GAAAAGUCAAGUACUUUAA[dT][dT]−3′), TrxR2 #3 (sense: 5′-CGUGGAACCUUCUCCCCAA[dT][dT]−3′). siRNAs were transfected with Lipofectamine® RNAiMAX transfection reagent (Invitrogen) using 20 nM siRNA in Opti-MEM with no antibiotic. After 48 h transfection, the cells were processed for different experiments or collected for protein extraction to detect knockdown efficiency.

To transiently transfect plasmids, cells were seeded at a density of $3 \times 10^6$ cells/dish in 10 cm dishes. A total of 10 μg indicated plasmid was introduced using the calcium phosphate transfection method and incubated for 24 h.

### Immunoblotting (IB)

Cells were washed with PBS and lysed in lysis buffer (10 mM Tris/HCl, 150 mM NaCl, 1 mM EDTA, 1 mM EGTA, 0.2% NP-40 and 0.5% Triton X-100, pH 7.5) supplemented with protease-inhibitor cocktail for 30 min on ice. The lysates were centrifugated at 20,000 × g for 10 min and the protein concentrations are measured using Bradford method. Equal amount of protein samples was separated using 11% or 4–20% SDS-PAGE and transferred onto PVDF membrane. Non-specific binding sites are blocked with 5% milk for 1 h at room temperature followed by incubation with the specific first antibody. After washing with TBST, membranes were then incubated for 1 h at room temperature with a peroxidase-conjugated secondary antibody. Membranes were again washed with TBST and then incubated with SuperSignal West Pico PLUS or SuperSignal West Femto Maximum Sensitivity chemiluminescent substrates (Thermo Fisher Scientific).

### Immunoprecipitation (IP)

Cells transfected with indicated plasmids or empty vector plasmid were incubated with DMSO (control) or AF for 2 h and lysed in cell lysis buffer (150 mM NaCl, 10 mM Tris/HCl, pH 7.4, 1 mM EDTA, 1 mM EGTA, 0.2% NP-40, and protease inhibitor mixture). Normally, 600 μg of total proteins were used for IP in a total volume of 700 μL. For ectopically expressed proteins, anti-FLAG M2 or anti-HA Affinity beads were incubated at 4 °C with equal lysate of cells for 2 h on a shaker. After removing the supernatant and washing with 700 μL of lysis buffer three times, the immunoprecipitates were processed for IB. Cells without transfection were prepared for IP for endogenous proteins.

### Analysis of protein ubiquitination in cells by re-immunoprecipitation (reIP)

NHK, Mcl1 or Mid49 ubiquitination was detected by reIP under denaturing condition as previously reported[69,81]. Briefly, cells stably expressing NHK-HA or transiently expressing Myc-Mcl1 or Myc-MiD49 were treated with 2 μM BTZ in the presence or absence of 100 nM AF for 4 h. After treatment, the cells were collected and lysed under native conditions followed by immunoprecipitation (IP) with anti-HA or anti-Myc antibody-conjugated beads for 2 h. To remove any protein that may associated with the tagged proteins, the immunoprecipitates were denatured with 2% SDS, and the beads were removed by centrifugation. The supernatants were then diluted in native lysis buffer from which NHK-HA or Myc-Mcl1 or Myc-MiD49 was re-immunoprecipitated followed by IB for HA and ubiquitin.

### Binding model prediction

The protein structure of UBA1 and UBE2G2 were retrieved from the Protein Data Bank (PDB code 6DC6 and 4LAD)[45,82]. The 3D structure of auranofin was obtained from the CSD (entry 1101703) (https://www.ccdc.cam.ac.uk/). Docking of auranofin was performed using the MOE dock program (www.chemcomp.com). Based on our MS study, the Au-triethylphosphine portion was involved in binding to UBA1, therefore, docking was performed only with the Au-Pt3 to the putative binding site to C1039. The ligand-induced fit protocol was applied and the binding affinity was evaluated using the GBVI/WSA score. The best binding model with the lowest binding free energies were selected and refined with MD simulations. The initial binding complex of UBA1 and E2G2 was predicted using the AlphaFold-Multimer program[83]. The binding complex of UBA1-Au(I)-Pt3-E2G2 was generated using the UBA1-E2G2 binding model as template and refined with MD simulations using the AMBER program[84].

### Mass spectrometry proteomics

**On-beads digestion.** IP samples in quadruplicates were digested using an on-beads digestion protocol. In brief, the beads were washed 5 times using 25 mM ammonium bicarbonate prior to adding 15 mM Tris (2-carboxyethyl) phosphine (TCEP), incubating for 30 min at 55 °C. After cooling down, 15 mM of iodoacetamide (IAA) was added to the beads and incubated in the dark for 30 min. This was followed by addition of 50 ng of trypsin to the beads and incubation at 37 °C overnight. The digestion was quenched by adding 1% of formic acid (FA) and the supernatant was transferred to a clean and protein lo-bind tube. The beads were resuspended in 60% acetonitrile (ACN) with 0.1% FA and incubated for 5 min, and then the second supernatant was transferred with the first supernatant. The samples were dried in the speed vac and resuspended in 15 μL of sample buffer (97.9% water, 2% ACN and 0.1% FA) prior to HPLC−MS/MS analysis. AF/UBA1 binding sites experiment: three samples were prepared including (1) NT, non-treated control UBA1, ~4.2 μM in PBS; (2) AF, 5 μM UBA1 incubated with ~21 μM AF at room temperature for 1 h; (3) AF low concentration (AF-LO), ~0.5 μM UBA1 incubated with ~2.5 μM AF at room temperature for 1 h. All the three samples were further denatured by adding 0.1% RapiGest prior to protein precipitation and trypsin/lys-C digestion at 37 °C overnight prior to HPLC−MS/MS analysis.

All proteomic HPLC−MS/MS analysis was performed using an UltiMate 3000-nano LC system coupled to the Orbitrap Fusion Lumos Tribrid mass spectrometer equipped with the Nanospray Flex ion source (Thermo Fisher). Peptides were loaded onto the trap column (Acclaim PepMap 100 C18, 75 μm × 2 cm, particle size: 3 μm, 100 Å) and separated with an analytical column with the spray tip (75 μm × 30 cm, 1.7 μm, 100 Å; CoAnn Technologies) using a 200 min method (~180 min gradient). Peptides were loaded onto the trap column by autosampler using loading solvent (2% acetonitrile in 98% UHPLC-grade water) at a flow rate of 4 μL/min. Elution of peptides from the analytical column

was performed using a 180 min gradient (including sample loading and re-equilibration) starting at 98% A (0.1% formic acid in UHPLC-grade water) at a flow rate of 300 nL/min. The mobile phase was maintained at 2% B (80% acetonitrile, 19.9% water, 0.1% formic acid) for 5 min, 2–9% B for 4 min, 9–38% B for 141 min, 38–50% B for 25 min, 50–90% B for 3 min, and maintained at 90% B for 10 min, followed by re-equilibration of the column with 2% B for 10 min. Column oven parameters were set as follows: temperature, 40 °C. For AF/UBA1 samples, a short gradient (45 min) method was used, the gradient was adjusted as below. The mobile phase was maintained at 2% B (80% acetonitrile, 19.9% water, 0.1% formic acid) for 5 min, 2–9% B for 2 min, 9–38% B for 33 min, 38–50% B for 2 min, 50–90% B for 3 min, and maintained at 90% B for 6 min, followed by re-equilibration of the column with 2% B for 9 min.

The mass spectrometer was operated in positive-ionization mode with the Nanospray Flex ion source with spray voltage set at 1800 V, and ion transfer tube temperature set at 250 °C. The MS scan was operated at data-dependent acquisition mode, with full MS scans over a mass range of m/z 375–1800 with detection in the Orbitrap (120 K resolution) and with auto gain control (AGC) set to $1.0 \times 10^6$. The fragment ion spectra were acquired in Orbitrap (15 K resolution) with a normalized collision energy of 28% at HCD activation mode. In each cycle of data-dependent acquisition analysis, the most intense ions above were selected for the MS/MS analysis, and the cycle time for MS and MS/MS analysis was set as 2 s. The AGC for MS/MS was set as Standard and a maximum injection time was 22 ms. Precursor ions with charges of +2 to +7 were isolated for MS/MS sequencing. The MS/MS isolation window was 1.2 Da, and the dynamic exclusion time was set at 60 s (after one MS/MS acquisition) with a mass tolerance of ±10 ppm.

**Data analysis.** Proteome Discoverer software suite (version 2.4, Thermo Fisher) with Sequest algorithm were used for peptide identification and quantitation. The MS raw data were searched against a Swiss-Prot human database (version Jan 2019, reviewed database) consisting of 20,350 entries using the following parameters: precursor ion mass tolerance of 10 ppm and a fragment ion mass tolerance of 0.02 Da. Peptides were searched using fully tryptic cleavage constraints and up to two internal cleavages sites were allowed for tryptic digestion. Fixed modifications consisted of carbamidomethylation of cysteine. Variable modifications considered were oxidation of methionine residues and N-terminal protein acetylation. Peptide identification false discovery rates (FDR) were limited to a maximum of 0.01 using identifications from a concatenated database from the non-decoy and the decoy databases. For UBA1 samples, the MS raw data were searched against a UBA1 protein database (Accession number: P22314) using the following parameters: precursor ion mass tolerance of 50 ppm and a fragment ion mass tolerance of 0.5 Da. Peptides were searched using fully tryptic cleavage constraints and up to four internal cleavages sites were allowed for tryptic digestion. Variable modifications considered were oxidation of methionine residues, phosphorylation (serine, threonine and tyrosine) and Auranofin (+314.0499 Da) of cysteine residue. Label-free quantification analysis used the "Precursor Ions Quantifier" node from Proteome Discoverer and normalized by total peptide amount.

## Surface plasmon resonance (SPR)
**Surface preparation.** Binding reactions was done in HBS-EP buffer from Biacore (Biacore Inc., New Jersey), containing 10 mM Hepes, 150 mM NaCl, 3 mM EDTA, and 0.05% (v/v) surfactant p20, pH 7.4. Solutions was filtered (0.2 μM) and degassed before use. Protein Wt, C1040A, and C1039A were coupled to the surface of a Biacore CM5 sensor chips flow cell-1, flow cell-2 and flow cell-3 respectively, by direct immobilization. The carboxymethyl-dextran surface of chip (flow cell 2, 3 and 4) was activated with a 35 μL injection of a mixture of 0.1 M NHS and 0.1 M EDC in water. An aliquot of 100 μL of 20 μg/mL Wt, C1040A, C1039A protein in 10 mM sodium acetate, pH 4.0, were

injected into flow cells 2, 3 and 4 of CM5 chip, to levels of 9000 resonance units (RU). The remaining NHS-ester active sites in the dextran surface flow cell-2 was blocked with 35 μl 1 M ethanolamine, pH 8.2, and washed at 50 μL/min with one pulse of 50 μL of 10 mM glycine pH 1.75 followed 50 μL of HBS-P. Flow cell-1 was used as reference, and was activated with 0.1 M NHS and 0.1 M EDC in water and blocked with 35 μL of 1 M ethanolamine, pH 8.2, without protein coupling.

**Kinetics analysis of binding.** In order to minimize mass transport effects, the binding analyses, small molecules AF was performed at flow rate of 30 μL/ min at 25°C. To avoid regeneration the single-cycle kinetics was performed. The analytes (60 μL each of analytes, 0-100 nM, in HBS-P buffer) were injected and the association was recorded by surface plasmon resonance (SPR) with a Biacore T200 (Cytiva, New Jersey). The signal from the blank channel (flow cell-1) was subtracted from the channel containing Wt, C1040A and C1039A protein.

**Data analysis.** Sensorgrams of the interaction generated by the instrument was analyzed using the Biacore T200 evaluation software version 3.4.2 (Cytiva Inc., New Jersey). The reference surface data were subtracted from the reaction surface data to correct for changes in the refractive index of the solution, injection noise and non-specific binding to the blank surface. A blank injection with buffer alone was subtracted from the resulting data. Data were globally fitted to the Lagmuir model for a 1:1 binding.

## Dislocation-induced reconstituted GFP (drGFP) assay[67]
HeLa cells stably expressing SP-S11-NHK-HA ($2 \times 10^4$ cells/well) were seeded in 96-well plate and cultured overnight. Then the medium was replaced by fresh medium before the cells were treated with various concentrations of AF and/or BTZ (0.2 μM) for 20 h. GFP images were acquired every 2 h under a 20 × objective lens using an IncuCyte S3 live-cell analysis system.

## Cycloheximide (CHX) chase assay[81]
HeLa cells were treated with cycloheximide (CHX, 50 μg/mL) alone or along with 100 nM AF for the indicated time. Aliquots of cells were collected, lysed and analyzed by IB.

## Cellular thermal shift assay (CETSA)[35]
Cells were treated with 2 μM AF for 1 h at 37 °C. After incubation, cells were harvested and lysed in lysis buffer (150 mM NaCl, 10 mM Tris/HCl, pH 7.4, 1 mM EDTA, 1 mM EGTA, 0.4% NP-40, and protease inhibitor mixture). The respective lysates were divided into smaller (50 μL) aliquots and heated individually at different temperatures for 3 min (Veriti Thermal Cycler, Applied Biosystems, Foster City, CA, USA) followed by cooling for 3 min at room temperature. The heated lysates were centrifuged at 20,000 g for 20 min at 4 °C to separate the soluble fractions from precipitates. The supernatants were transferred to new tubes and analyzed by SDS-PAGE followed by IB.

## MBP-pull down assay
MBP fusion constructs of UBA1 (1-220 or AAD-UFD, 5 μg) immobilized on MBP-Sepharose beads were treated with indicated concentrations of AF for 1.5 h. After washing for three times at 2200 × g for 2 min, the beads were incubated with the indicated recombinant E2 in lysis buffer (150 mM NaCl, 10 mM Tris/HCl, pH 7.4, 1 mM EDTA, 1 mM EGTA and 0.2% NP-40) for another 2 h at room temperature. After washing, beads were processed for SDS-PAGE and IB. MBP-UBA1 fragments were detected by Ponceau S staining.

## In vitro autoubiquitination assay
In vitro ubiquitination assay has been reported[76,77,85]. Briefly, GST-E3 was immobilized on glutathione beads. The ubiquitination reaction

mix contained 25 nM UBA1, 125 nM His-E2, 400 ng GST-E3, 4 μg ubiquitin and 12.5 nM AF in 20 μL reaction buffers (50 mM Tris−HCl, pH 7.5, 5 mM $MgCl_2$, 2 mM ATP and 2 mM DTT). The reaction mix was incubated for 7.5 or 15 min at 37 °C and halted by adding 7 μL loading buffer followed by processing for immunoblotting. The negative controls included reactions without E1.

### Ubiquitin charging (trans-thioesterification) assays
UBA1 charging assay: 250 nM of recombinant UBA1 was pre-treated with different doses of AF (0, 12.5, 25, 50 or 100 nM) in reaction buffer (5 mM Tris−HCl, pH 7.5, 0.5 mM $MgCl_2$, 0.5 mM KCl) for 1 min at room temperature followed by adding 50 μM ATP and incubated at 15 °C for 45 s. Reactions were stopped by adding non-reducing loading buffer.

E2 charging assay: 250 nM UBA1 and 4 μM His-E2s (His-UBE2G2 or His-UBE2D1) were treated with increasing concentrations of AF (0, 6.25, 12.5, 25 or 50 nM) in reaction buffer (5 mM Tris−HCl, pH 7.5, 0.5 mM $MgCl_2$, 0.5 mM KCl) for 1 min at room temperature. ATP (50 μM) was added to the reactions as indicated. The reactions were incubated at 15 °C for 45 s and stopped by adding non-reducing loading buffer. In control reactions, 5 mM DTT was used to disrupt the thioester bond that links ubiquitin to the active cysteine of E2s.

### CRISPR genome editing
A Cas9 nickase (D10A) based PAM-out strategy was used to create C1039A mutation in the genomic UBA1 gene of HCT116 cells. Two sgRNAs targeting different strands close to the mutation site and the single-stranded oligodeoxynucleotide (ssODN) donor were designed by Alt-R™ CRISPR HDR Design Tool (www.idtdna.com) (Supplementary Table 2). The ssODN donor contains the mutated codon for C1039A mutation and some silent mutations in the corresponding regions of the sgRNAs' targeting sequences. The target DNA sequences of the sgRNAs were cloned into pSpCas9n(BB)−2A-Puro (PX462) V2.0 (Addgene plasmid #62987)[86]. HCT116 cells were transfected with both sgRNA constructs and the ssODN donor using lipofectamine 3000 (Invitrogen). Transfected cells were selected by puromycin (2 μg/mL) for 3 days. Single clones were screened by polymerase chain reaction (PCR) using primer pairs specific to the mutation sequence. PCR positive clones were further verified by DNA sequencing of the mutated region.

### Quantitative RT-PCR (qPCR)
Total RNA was prepared using Trizol reagent (Invitrogen) according to the manufacturer's protocol. Reverse transcription was performed using ProtoScript® First Strand cDNA Synthesis Kit (New England BioLabs) using 2 μg total RNA/reaction. Primers used in qPCR were listed below. Sense primer for NHK: 5′-ATGCCGTCTTCTGTCTCGTGG −3′; antisense primer for NHK: 5′-GCACGGCCTTGGAGAGCTTC-3′, sense primer for CD3δ: 5′-TGTAATGGGACAGAGCAGCTG-3′; antisense primer for CD3δ: 5′-TTATGCGTAGTCTGGGACGTCG-3′; sense primer for β-actin: 5′-CACCAACTGGGACGACAT-3′; antisense primer for β-actin: 5′-ACAGCCTGGATAGCAACG-3′. To assess NHK and CD3δ expression level treated by different doses of AF, qPCR was performed using a CFX96 Touch Real-Time PCR Detection System (Bio-rad). The qPCR reactions were set up for detection of NHK, CD3δ or β-actin in a total volume of 30 μl using iQ™ SYBR® Green Supermix (Bio-rad). Primer sets we used for NHK, CD3δ or β-actin have similar amplification efficiencies. To calculate the relative NHK, CD3δ mRNA levels for each sample, the threshold cycle (Ct) value was normalized to the value for β-actin (ΔCt = Ct (NHK or CD3δ) − Ct (β-actin)). The relative NHK or CD3δ mRNA levels were calculated as 2ΔCt for each sample.

### Statistical analysis and reproducibility
All quantitative data presented are representative of at least three independent experiments. Quantifications are shown as means ± S.D.

Differences between two groups were analyzed by paired or unpaired two-tailed Student's $t$-test using GraphPad Prism 7.0. For all analyses, the $p$-value was considered significant as follows: *$p < 0.05$, **$p < 0.01$, ***$p < 0.001$. A representative immunoblot from three biological repeats with similar results is shown in Figs. 2a, 3c−g, 4a−d, 5a−c, f, 7a−d, and supplementary Figs. 1a, b, 5, 6a−f, 8a−c. Raw data and uncropped/replicate blots are provided as a Source data file.

## Data availability
Source data are provided with this paper. The mass spectrometry proteomics data have been deposited to the ProteomeXchange Consortium via the PRIDE[87] partner repository with the dataset identifier PXD042558. The MS raw data were searched against a Swiss-Prot human database (version Jan 2019, reviewed database). For UBA1 samples, the MS raw data were searched against a UBA1 protein database (Accession number: P22314). Source data are provided with this paper.

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

## Acknowledgements

We thank Dr. Allan M. Weissman for his invaluable suggestions during this study, Drs. David B. Beck and Achim Warner for their helpful discussions and for kindly providing reagents, and Drs. Angelos Constantinou and Mariusz Karbowski for gifts of plasmid constructs. S.F. was funded by the National Institute of Arthritis and Musculoskeletal and Skin Diseases (NIAMS) (UO1AR081599-01), National Institutes of Health (NIH); X.H., T.X., S.K., B.B., Y.Q., D.C.T., C.A.L., A.S., R.H., G.R., M.J.H., and D.T. were supported by the intramural research program, National Center for Advancing Translational Sciences (NCATS), NIH; Y.Y. was supported by the intramural research program, the National Institute of Diabetes and Digestive and Kidney Diseases (NIDDK), NIH.

## Author contributions

S.F. conceived the idea. S.F., D.T. and W.J. developed hypotheses. W.J. conducted most experiments. Y.Z. generated C1039A cells and participated in experiments. X.H. conducted modeling study. T. X. performed bioinformatics analysis with R.H. Y. Zhang conducted SPR experiments. S.K. B.B. and D.C.T. participated in biochemical experiments. D.T. performed mass spectrometry studies with Y.Q. and C.A.L. D.T., A.S., S.F., M.J.H., G.R., Y.Y., and B.M.P. participated in experimental design and supervised the study. S.F. and D.T. wrote the manuscript with M.J.H., D.R., and all authors approved the final version.

## Competing interests

The authors declare no competing interests.
