## [Peer Review File · Nature Communications]

REVIEWER COMMENTS

Reviewer #1 (Remarks to the Author):

Yan et al. report that the organogold compound Auranofin (AF) acts as an activator of the human ubiquitin activating enzyme UBA1 as demonstrated by increased transfer of ubiquitin onto the majority of ubiquitin conjugating enzymes, increased autoubiquitylation levels of 5 ubiquitin ligases and reduced levels of a few tested substrate proteins. As this would be the first UBA1 activator this discovery is of considerable interest to the ubiquitin community, in particular as such a compound could be ultimately of therapeutic interest. In the present form, however, major revisions are required to render this manuscript acceptable.

Major points:

1. The Biacore experiments need to be improved in several ways. (a) The description in Materials and Methods is inadequate, e.g., no information is given on how the data were fit. The minimal text provided will not allow someone to reproduce the data. (b) In light of the predicted covalent binding, can this even be analyzed in a rigorous manner by SPR? The authors should provide examples where covalent ligand binding was successfully analyzed by SPR. They should also indicate how often the experiment was repeated. Right now my conclusion would be that $n=1$. (c) The fit (presumably in red?) fails to reproduce the dissociation phase in a non-systematic manner, i.e. it is higher at lower AF concentrations, about correct for the central concentration, lower for concentration 4 and both above and below the data points at the highest concentration. (d) Suitable controls are missing, in particular SPR experiments need to be performed with the C1039A variant as a negative control and a control in the absence of UBA1 to analyze the binding of AF to the sensor. (e) The authors report a dissociation constant of 5.19 nM with an absurd precision of two digits after the decimal point in the absence of any experimental error.

2. The docking experiments represent another weak point of the manuscript. Presumably guided by the crystal structure of trypanothione reductase (PDB entry 2YAU) in complex with AF in which it appears that the Au-S bond in AF was broken upon ligand binding, the authors embarked on a two-step protocol in which the Au atom was first docked to C1039 followed by docking of the organic moiety to the UFD. Relying on this particular PDB entry is a terrible idea for two reasons: It describes a low resolution structure (3.5 Å), which is not even well refined. More importantly, inspection of the electron density maps indicates that the ligand, without a doubt, was modeled incorrectly. It is also not a good idea to derive the coordinates of AF from such a low resolution structure. Have the authors checked the Cambridge Small Molecule Database for an AF entry? The authors should also ask themselves where specificity would originate in their two-step approach; if the Au-atom can dissociate so easily, it should be able to attack any highly solvent accessible cysteine as long as there are no clashes with the organic moiety. In particular the active site cysteine of UBA1 should be able to react with AF. Another puzzling aspect of the docking model appears to be that the organic moiety of AF in fact apparently blocks the common E2 binding site on the ubiquitin fold domain (UFD) of UBA1. Have the authors analyzed their docking pose in the framework of the available E1-E2 complexes? If AF indeed acts as a molecular glue, as suggested by the authors, AF should engage in productive interactions with both UBA1 and the bound E2. Regarding the L954A and E1037A mutants which abrogate the AF-mediated effect, the authors need to rule out that these UBA1 variants do not lead to either (local) structural changes or altered enzyme stability.

3. The data identifying C1039 as possible target site for AF (Fig. 2A) are not convincing. The UBE2G2 signal for C1039A in the absence of AF is very weak (and possibly affected by a spot on the right hand side of the band). Ignoring this artefact there is a clear increase of the signal in the absence of AF vs. the presence of AF. Also for the C1039A and C1040A variants the authors need to demonstrate that these mutations do not affect the folding/stability of UBA1.

4. The data in Fig. 6C are not convincing since the band intensity in the absence and presence of AF is not that strong. Maybe quantification would paint a clearer picture.

Minor points:

1. The authors should consider replacing "transthiolation", i.e. the exchange of SH-groups, with the chemically correct term "transthioesterification".
2. Since the authors decided to list the individual classes of E3 ligases in the introduction, they should also mention RBR ligases.
3. Please define "drGFP".
4. Please define "AUP1".
5. Please comment on the strong MonoUb signal in Figure 4, panel B. Is this an artefact or does AF induce expression of ubiquitin?
6. In Fig. 6A, there is a signal above the Mcl1 band. Please comment on its identity.

Reviewer #2 (Remarks to the Author):

In the manuscript Yan et al, the authors describe the identification of a previously unknown activity of the molecule Auranofin (AF) to covalently bind to UBA1 and alter its affinity for E2 enzymes. The authors claim that AF binds to the ubiquitin fold domain of UBA1 and enhances the recruitment across the family of E2 enzymes resulting in enhanced trans-thiolation and ultimately downstream E3 driven target ubiquitylation and substrate turnover in cells.

The ubiquitin-like protein field has benefitted from the discovery of useful inhibitors /modulators of the proteasome, E1, E2 and E3 enzymes. It is critical that studies clearly demonstrate MoA of novel molecules and discern the contribution of on target, on mechanism activities from non-specific effects.

Auranofin is a reactive molecule that has been reported to induce a range of effect from oxidative stress to UPR/ER stress induction thus there is a risk of non-specific effects contributing to a range of phenomenology as the authors openly describe and which is greatly appreciated by this reviewer.

It is interesting to consider that AF may be a bona fide UBA1 enhancer. The authors utilized several compelling approaches (CETSA, site specific mutation, coIPs and other assays) to build a weight of evidence that AF may be enhancing E1:E2 interactions, however, given the nonspecific effects of this compound there are further studies warranted to confirm the mechanism of action. In general, in addition to the studies provided, greater use of LC/MS based approaches to confirm UBA1 modification as well as use of point mutations to confirm AF-UBA1 modification is responsible for the effects is warranted (details below)

Comments on Figures:

Figures1/2:

- Have the authors confirmed AF forms covalent modification on the UFD of UBA1? It is important to demonstrate/determine if AF is covalently reacting with the UFD cysteine by LC/MS. This is suggested by multiple experiments including the experiment in Fig S1 and the work in Figure 2 but isn't definitively proven. Why?
 - a. If it is covalent, also demonstrate that preincubated UBA1-AF adduct retains the enhanced E2 interaction activity even when free AF is purified away. This is shown in the experiment in FigS1 for UFD but what about full length UBA1? Did the authors attempt this with full length UBA1?
 - b. Utilize LC/MS to demonstrate that other proteins in coIPs or assay reactions are not being modified (E3, E2, etc).
 - c. Request: the 5.19nM Kd (with 2 sig digits) is quite specific and is an important point. This reviewer is not an SPR expert, but is it possible to calculate a kD from a covalent binder at a level

that is lower than the lowest measured concentration? What is the K_D with the C1039A, E1037A and L954A?

Figure 2:

- Great to see the exploration and use of the point mutations to evaluate on mechanism/on target AF activity. These mutants are powerful tools, provided they do not affect UBA1 activity, and should be utilized throughout the experiments in other figures.
- Fig 2E: blocking with IAA indicates it could be covalent but is not definitive proof that AF is modifying UBA1. Did the authors attempt LC/MS confirmation to ID the covalently modified peptide? This would be a critical result as mentioned above.

Figure 3:

- This reviewer appreciates the intent of the coIP LC/MS study and appreciate the thoughtfulness of including the Sumo and UBA6/Ub pairings, but the statements require the C1039 pt mutants as important controls (if proven to be the covalent modification site). Showing that the C1039A point mutation fully abrogates the AF induced coIP phenomenon would greatly strengthen the claims of this manuscript.

Figure 4:

- Making an argument about enhanced E2 trans-thiolation is tricky given how efficient these reactions often are. In the absence of AF, the E2 trans-thiolation shown appears very weak – this reaction is usually quite robust. Did the authors perform a time course or other kinetic experiments?
- Was the experiment tested with pretreated and purified UBA1-AF to confirm any enhancement was through the mechanism the authors claim and not AF reacting with other proteins in the assay mixture?

Figure 5:

- Pleased to see the use of in vitro E3 auto ubiquitination assays.
- Missing control of compound binding refractory mutants in the assay.
- Please explain:
 - o why so little autoubiquitination of the E3s is shown in the absence of AF. Is this a blot exposure issue?
 - o Some E3s are low MW, others like E6AP are ~100kDa yet the poly Ub is mostly below the 95kDa marker suggesting free Ub chains or other proteins are being modified. Have the authors confirmed this is autoubiquitination of the actual E3?
- Recommend a time course be shown for at least 1 E2/E3 pair.

Figure 6:

- AF is pleiotropic, need UBA1 compound binding point mutant controls to confirm that AF effects on protein turnover are through the UBA1-AF/E2 mechanism being claimed:
 - o Recommended experiment: Engineer compound refractory pt mutations into UBA1 in cells (using CRISPR) and demonstrate that the effects of the compound on enhanced protein turnover are dependent on the UBA1 modification. As mentioned in the publication, AF is a known inducer of oxidative stress, ER stress and UPR – which could lead to effects on protein translation, chaperone induction, etc.
- Did the authors perform a covalent modification scan (e.g. like what D. Nomura or B. Cravatt do) to see which proteins are modified in the cell? What if deubiquitylating enzymes such as the thiol dependent ubiquitin specific proteases are being modified... this could enhance protein turnover too through an unrelated mechanism.

General comment:

The rate limiting step of protein ubiquitylation is at the level of the substrate/E3 protein, not typically at the level of the E1 activating enzyme. This manuscript would infer that E2-Ub thioester levels are limiting, which is to this reviewer's knowledge has not been shown to be the case.

Minor points:

- Should always include the chemical structure in a manuscript.

- First sentence of abstract: "UBA1 is the primary E1 ubiquitin-activating enzyme, regulating stability and function of numerous proteins via initiating their ubiquitination". Recommend rewording. The ubiquitin proteasome system (UPS) not UBA1 per se regulates the stability and functions of cellular proteins. UBA1 ensures that sufficient activated ubiquitin is available for the UPS to perform its cellular function.

Reviewer #3 (Remarks to the Author):

Small molecules targeting ubiquitin-mediated proteasomal degradation pathway remain poorly defined. In this manuscript, by using in vitro assays, Yan et al verify auranofin (AF), a drug approved for the treatment of rheumatoid arthritis, is an enhancer specifically for the E1 ubiquitin-activating enzyme UBA1 and facilitates ERAD as well as outer mitochondrial membrane-associated degradation. The study is overall very interesting and solid, and is well suited to be published in NC.

However, some conclusions are not clear and robust, and some additional key experiments may be needed to make the study more comprehensive and impactful.

Major points:

The major conclusion of this paper is that AF promotes protein ubiquitination and degradation in cells, including the ERAD process. The authors demonstrated this using the model ERAD substrates NHK-HA and CD3d-HA (Fig. 5D-5H). However, the data is not very convincing with several concerns are: First, UBA1 is not known to be involved in ERAD. Although the Fig 5C showed that UBA1 can enhance the autoubiquitination of ERAD E3 gp78, it is not sufficient to conclude that E3 activity is enhanced. It is critical to show that UBA1 is required for ERAD by generating the UBA1 KO cells to examine whether known ERAD substrates (OS9, IRE1a and CD147) are stabilized and accumulate. Moreover, HRD1 is a much better known E3 ligase involved in ERAD, hence autoubiquitination assay should also be performed for HRD1, not just gp78.

Some experiments require important controls: The mRNA levels should be shown in Fig 5D and 5H. Fig 6A should test NHK-HA and CD3d-HA. Fig 5A should include one negative control, for example, changing E1 (UBA1) to UBA6 which is not affected by AF, and should show total protein levels of E1, E2 and E3. Fig 1C-D is quite confusing and more explanation and description are needed - What do these lines mean and why do two lines look comparable in Fig 1C? For Fig 1E, UBA1 KO 293T should be included as a control to make the conclusion more solid. For Fig 3A, please explain for the first two panels. For Fig 3C, it would be better to confirm the interactions using endogenous IP of anti-UBA1 antibody as shown in Fig. 1A.

Minor Points:

- 1) The rationale for testing AF is missing.
- 1) Show the domain structure for UBA1 in Fig 2.
- 2) Which cell type was used in Fig 2A? The cell type of 293T cells was shown in the text but HeLa cell was shown in the figure legend.
- 3) Which cell type was used in Fig 2D? No mention in both text and figure legend.
- 4) Statistic analysis for Fig 5G.
- 5) What is the E1 in Fig 5A? Please mention it in the figure and the figure legend.

We would like to thank the reviewers for providing positive assessments of our manuscript and for offering valid concerns and valuable suggestions for improvement. As you will see, we have included a significant amount of new data to address the reviewers' concerns and have made a comprehensive revision of the manuscript. The following are our point-to-point responses (in blue):

REVIEWER COMMENTS

Reviewer #1 (Remarks to the Author):

Yan et al. report that the organogold compound Auranofin (AF) acts as an activator of the human ubiquitin activating enzyme UBA1 as demonstrated by increased transfer of ubiquitin onto the majority of ubiquitin conjugating enzymes, increased autoubiquitylation levels of 5 ubiquitin ligases and reduced levels of a few tested substrate proteins. As this would be the first UBA1 activator this discovery is of considerable interest to the ubiquitin community, in particular as such a compound could be ultimately of therapeutic interest. In the present form, however, major revisions are required to render this manuscript acceptable.

We would like to thank the reviewer for providing valuable comments regarding the significance of our discovery. The reviewer also raised valid and significant concerns, which have helped us tremendously in improving this revised manuscript.

Major points:

1. The Biacore experiments need to be improved in several ways. (a) The description in Materials and Methods is inadequate, e.g., no information is given on how the data were fit. The minimal text provided will not allow someone to reproduce the data.

Response: The detailed methods for Biacore experiments have been provided in the Materials and Methods section.

(b) In light of the predicted covalent binding, can this even be analyzed in a rigorous manner by SPR? The authors should provide examples where covalent ligand binding was successfully analyzed by SPR. They should also indicate how often the experiment was repeated. Right now my conclusion would be that $n=1$.

Response: We appreciate the reviewer's concern regarding the rigorous analysis of covalent binding by SPR. As suggested, we have repeated the SPR experiments with wt UBA1 (now presented in Supplementary Fig. 3A) and added new data on UBA1 mutants (Supplementary Fig. 3C & D: C1039A and C1040A). The results are consistent with our proposed AF-UBA1 binding model in which C1039 but not C1040 is critically required for AF binding.

SPR can be used to assess binding of covalent ligands, albeit sometimes it is not employed because covalent inhibitors can preclude reconstitution of the sensor chip, preventing subsequent usage. For this reason, we performed the experiment in single-cycle format, in which ligand concentration is increased sequentially. An example of single-cycle SPR characterization of a covalent ligand is reported in Mastraccio et al, ACS Med Chem Lett 2021; (<https://pubs.acs.org/doi/10.1021/acsmmedchemlett.0c00654>). Another report of SPR for a covalent BMX inhibitor was reported by Seixas et al, RSC Chem Biol 2020 (<https://pubs.rsc.org/en/content/articlehtml/2020/cb/d0cb00033g>). Notably, the dissociation phase can be difficult to fit because it is dependent on non-covalent k_{off} and also the k_{inact} (rate constant of covalent and irreversible inactivation), and distortion to the binding/dissociation curves has been proposed to be informative towards calculating the k_{inact} (<https://www.sartorius.co.kr/wp-content/page/download/552364/kinetics-of-irreversible-inhibitors-on-pioneer-fe-system-application-note-en-sartorius-data.pdf>).

Because of the complexities of interpreting binding/dissociation curves for covalent ligands and the imprecise fit of the data, we agree that the calculated K_d is not of highest confidence, and therefore have removed those values from the plots. We do think, however, that the SPR data clearly demonstrates binding for the WT and C1040A mutants, but not C1039A, and have therefore left it in the supplement as evidence of binding (Supplementary Fig. 3). The SPR data is not presented as the sole line of evidence for direct binding of AF to UBA1, but rather complements other experimental evidence, as outlined below.

We have now validated the covalent binding of AF to UBA1 by HPLC-MS/MS, in which a mass shift consistent with AF conjugated to Cys1039/1040 was detected. (Fig, S2). Moreover, we have also generated a HCT116 cell line harboring a C1039 mutation by CRISPR/Cas9 technology and showed that AF is no longer able to promote protein ubiquitination and degradation in the C1039 mutant cells (Fig. 6). These two experiments provide strong evidence that Cys1039 is the target of covalent modification by AF and that Uba1 targeting is responsible for cell-based effects described in our experiments.

(c) The fit (presumably in red?) fails to reproduce the dissociation phase in a non-systematic manner, i.e. it is higher at lower AF concentrations, about correct for the central concentration, lower for concentration 4 and both above and below the data points at the highest concentration.

Response: Thank you for pointing out the deficiency of this dataset. We have repeated the SPR experiment and the new data is presented in Supplementary Fig. 3. While the K_D values are similar to the initial results, we are no longer reporting the K_D in light of challenges with fitting the experimental data.

(d) Suitable controls are missing, in particular SPR experiments need to be performed with the C1039A variant as a negative control and a control in the absence of UBA1 to analyze the binding of AF to the sensor.

Response: C1039A as a negative control and C1040A as a positive control have now been added (Supplementary Fig. 3A, B). In our first submission, we have shown that AF does not bind to the sensor coated with UBE2G2 and now we show that AF does not bind to sensor coated with UBA1C1039A Supplementary (Fig. 3B). Therefore, we are confident that AF does not bind to the sensor.

(e) The authors report a dissociation constant of 5.19 nM with an absurd precision of two digits after the decimal point in the absence of any experimental error.

Response: Thank you for bringing this to our attention. We agree with the reviewer's concern regarding the precision of the dissociation constant. We have now removed the Kd metric and revised the text accordingly. The SPR data is shown just as evidence of AF-UBA1 binding.

2. The docking experiments represent another weak point of the manuscript. Presumably guided by the crystal structure of trypanothione reductase (PDB entry 2YAU) in complex with AF in which it appears that the Au-S bond in AF was broken upon ligand binding, the authors embarked on a two-step protocol in which the Au atom was first docked to C1039 followed by docking of the organic moiety to the UFD. Relying on this particular PDB entry is a terrible idea for two reasons: It describes a low resolution structure (3.5 Å), which is not even well refined. More importantly, inspection of the electron density maps indicates that the ligand, without a doubt, was modeled incorrectly. It is also not a good idea to derive the coordinates of AF from such a low resolution structure. Have the authors checked the Cambridge Small Molecule Database for an AF entry? The authors should also ask themselves where specificity would originate in their two-step approach; if the Au-atom can dissociate so easily, it should be able to attack any highly solvent accessible cysteine as long as there are no clashes with the organic moiety. In particular the active site cysteine of UBA1 should be able to react with AF. Another puzzling aspect of the docking model appears to be that the organic moiety of AF in fact apparently blocks the common E2 binding site on the ubiquitin fold domain (UFD) of UBA1. Have the authors analyzed their docking pose in the framework of the available E1-E2 complexes? If AF indeed acts as a molecular glue, as suggested by the authors, AF should engage in productive interactions with both UBA1 and the bound E2. Regarding the L954A and E1037A mutants which abrogate the AF-mediated effect, the authors need to rule out that these UBA1 variants do not lead to either (local) structural changes or altered enzyme stability.

Response: There are two major concerns on docking experiments.

1) We agree that the reported crystal structure of 2YAU is not a good template for UBA1/AF docking due to its low resolution and poor quality. As suggested, we retrieved the crystal structure of AF deposited in CSD (entry 1101703) and used as a starting point for UBA1 modeling and docking in this study. Our new HPLC-MS/MS data and biochemical studies showed that AF bound to UBA1 through Au-PET3 by forming an adduct S-Au-PET3 with the thiol group of Cys1039 (Supplementary Fig 2, and Fig. 2, 3E, 7). The same reaction mechanism of AF with protein target has also been previously reported (Pratesi et al, *Inorg. Chem.* 2018, 57, 10507, DOI: 10.1021/acs.inorgchem.8b02177; Zoppi et al, *Dalton Trans.*, 2020, 49, 5906, DOI: 10.1039/d0dt00283f).

Based on the experimental finding, we revised our modeling approach, docked the Au-PET3 moiety of AF to the Cys1039 binding site (Fig. 2D, also see on the right). Moreover, we modeled the binding complex of UBA1 with UBE2G2 using AlphaFold2-multimer and protein-protein docking.

Thank for the reviewer's suggestion! Our revised model indeed show AF binds to both UBA1 and UBE2G2. The predicted binding model of UBA1-Au-PET3-UBE2G2 is shown in Fig. 2D and discussed in the revised manuscript.

Regarding the conjugation of AF to the active cysteine of UBA1, our data strongly indicate that it does not occur. In our HPLC-MS/MS study, we observed that only one tryptic peptide (aa1025 to 1054), which contains C1039 and C1040, was conjugated with a single AF molecule (Supplementary Fig. 2). This finding suggests that AF does not conjugate to the active cysteine of UBA1. Furthermore, if AF were to conjugate to the active cysteine of UBA1, it would result in the inactivation of UBA1's ability to charge ubiquitin to E2s and impair the ubiquitination catalyzed by E3s. However, to the contrary, our data show that AF actually enhances ubiquitin charging to E2s and promotes the activities of E3 enzymes.

2) Potential problem associated with L954A and E1037A mutants.

We appreciate the reviewer's concern regarding the potential effects of mutations on UBA1 folding/activity. We have indeed observed that the L954A mutation reduces UBA1 activity in catalyzing Hrd1c-mediated ubiquitination in vitro, as illustrated in the figure below.

The mechanism of the L954A mutation effects on UBA1 activity is unclear. Based on our new AF binding model, as shown in the right, L954 is located at the bottom of the binding pocket and does not make any interactions directly with AF. However, L954 forms hydrophobic interactions with V589 and W955 at the UFD-AAD hinge region, which likely plays an important role on protein stability and conformational changes upon AF or E2 binding. The abrogation of UBA1 activity with L954A mutant might be a result of local structural changes or enzyme stability caused by disruption of the interactions.

However, we cannot definitively conclude the exact binding mechanism of L954A mutation underlying AF-UBA1-E2 association. We believe that further structural studies are ultimately necessary to elucidate the effect of this mutation. In the revised manuscript we did not address the L954A mutant in the context of predicted binding model. Instead, we have demonstrated that mutations in two key predicted binding residues, E1037 and E1049, diminished the effect of AF. These results are intended to provide preliminary data supporting our proposed model.

3. The data identifying C1039 as possible target site for AF (Fig. 2A) are not convincing. The UBE2G2 signal for C1039A in the absence of AF is very weak (and possibly affected by a spot on the right hand side of the band). Ignoring this artefact there is a clear increase of the signal in the absence of AF vs. the presence of AF. Also for the C1039A and C1040A variants the authors need to demonstrate that these mutations do not affect the folding/stability of UBA1.

Response: Thank you for raising this concern. We agree with the reviewer's comment and have revised our manuscript accordingly. We have now used new data that replaced Fig. 2A (now Fig. 2C in this revised version). Additionally, we have provided additional evidence to support C1039 as a likely target site for AF, including the identification of C1039/1040 as a site of AF conjugation by mass spectrometry (Supplementary Fig. 2) and the creation of C1039A mutant HCT116 cells by CRISPR/cas9 in which the mutation reduced the effect of AF on the ubiquitination

and degradation of NHK in cells (Fig. 7). We believe that these new data provide strong support for the involvement of C1039 as a target site for AF.

To address the concern over the effects of C1039A and C1040A mutations on their folding/stability, we have now performed additional experiments. Specifically, we have assessed the effects of these mutations on UBA1 activity in ubiquitin activation (ubiquitin charging to UBA1) and showed that the mutations do not affect UBA1 activity in ubiquitin activation (Supplementary Fig. 6F). This new data suggests that C1039A and C1040A mutations do not significantly affect folding/stability of UBA1.

4. The data in Fig. 6C are not convincing since the band intensity in the absence and presence of AF is not that strong. Maybe quantification would paint a clearer picture.

Response: Thank you for bringing up this issue. We have included a new data from a replicate experiment, and the new data shows a clearer difference in band intensity. Please see Fig. 6C.

Minor points:

1. The authors should consider replacing "transthioation", i.e. the exchange of SH-groups, with the chemically correct term "transthioesterification".

Response: We agree with the reviewer. Revisions have been made in the title and text as suggested.

2. Since the authors decided to list the individual classes of E3 ligases in the introduction, they should also mention RBR ligases.

Response: As suggested, parkin as a representative E3 of RBR ligases has been tested and the data is in Supplementary Fig. 6C, showing that auranofin enhances its E3 activity in vitro. We have added text to the manuscript.

3. Please define "drGFP".

Response: drGFP is the short form of "dislocation-induced reconstituted GFP" and has now been defined in this revised manuscript.

4. Please define "AUP1".

Response: AUP1 is the short form of "ancient ubiquitous protein 1" and has now been defined in this revised manuscript.

5. Please comment on the strong MonoUb signal in Figure 4, panel B. Is this an artefact or does AF induce expression of ubiquitin?

Response: Thank you for bringing this to our attention. We have re-evaluated the data and found that the strong MonoUb signal in the last lane of Fig. 4B is an artifact due to contamination. We have added a short-exposed blot to show the contamination, and we apologize for any confusion caused. AF cannot induce expression of ubiquitin because it is an in vitro reconstitution assay.

6. In Fig. 6A, there is a signal above the Mcl1 band. Please comment on its identity.

Response: Thank you for pointing this out. We apologize for the confusion. The signal above the Mcl1 band in Fig. 6A was due to non-specific staining of a band in the protein molecular weight marker, which was not denoted in the figure legend. We have included a new data from a repeat experiment in Fig. 6A without the molecular weight marker, clearly demonstrating that the band is not related to Mcl1 or any other protein in the sample.

Reviewer #2 (Remarks to the Author):

In the manuscript Yan et al, the authors describe the identification of a previously unknown activity of the molecule Auranofin (AF) to covalently bind to UBA1 and alter its affinity for E2 enzymes. The authors claim that AF binds to the ubiquitin fold domain of UBA1 and enhances the recruitment across the family of E2 enzymes resulting in enhanced trans-thiolation and ultimately downstream E3 driven target ubiquitylation and substrate turnover in cells.

The ubiquitin-like protein field has benefitted from the discovery of useful inhibitors /modulators of the proteasome, E1, E2 and E3 enzymes. It is critical that studies clearly demonstrate MoA of novel molecules and discern the contribution of on target, on mechanism activities from non-specific effects.

Auranofin is a reactive molecule that has been reported to induce a range of effect from oxidative stress to UPR/ER stress induction thus there is a risk of non-specific effects contributing to a range of phenomenology as the authors openly describe and which is greatly appreciated by this reviewer.

It is interesting to consider that AF may be a bona fide UBA1 enhancer. The authors utilized several compelling approaches (CETSA, site specific mutation, coIPs and other assays) to build a weight of evidence that AF may be enhancing E1:E2 interactions, however, given the nonspecific effects of this compound there are further studies warranted to confirm the mechanism of action. In general, in addition to the studies provided, greater use of LC/MS based approaches to confirm UBA1 modification as well as use of point mutations to confirm AF-UBA1 modification is responsible for the effects is warranted (details below)

We would like to express our gratitude to the reviewer for the enthusiasm regarding our discovery. Additionally, we appreciate the reviewer for bringing up valid and significant concerns, which have greatly assisted us in enhancing this revised manuscript.

Comments on Figures:

Figures1/2:

- Have the authors confirmed AF forms covalent modification on the UFD of UBA1? It is important to demonstrate/determine if AF is covalently reacting with the UFD cysteine by LC/MS. This is suggested by multiple experiments including the experiment in Fig S1 and the work in Figure 2 but isn't definitively proven. Why?

Response: Yes, we have now performed HPLC-MS/MS experiments and confirmed that AF covalently modify C1039 in the UFD of UBA1 in both the isolated UFD and the full-length UBA1. The result is presented in Supplementary Fig. 2. To identify the potential AF modification sites by HPLC-MS/MS, we prepared three samples, 1) NT, non-treated control UBA1, ~ 4.2 μ M in PBS; 2) AF, 5 μ M UBA1 incubated with ~21 μ M AF at room temperature for 1 hour; 3) AF low concentration (AF-LO), ~0.5 μ M UBA1 incubated with ~2.5 μ M AF at room temperature for 1 hour. All three samples were denatured (no reduction and alkylation) and digested prior to HPLC-MS/MS analysis. After database search with AF modification as dynamic modification along with Oxidation (M) and Phosphorylation (S,T,Y), the UBA1 protein was identified with a sequence coverage at 90.83%. Importantly, we found AF (C) modifications on either C1039 or C1040. With further check, there was only one MS/MS spectra from samples AF and AF-LO corresponding to the peptide sequence "RKLGRHVRALVLELCCNDESGEDVEVPYVR" with an AF (+314.0499 Da) on either C1039 or C1040, shown as Supplementary Fig. 2 in the revised manuscript. No MS/MS spectra was found from NT sample matching to this sequence. Since no ions (b or y) were found between C1039 and C1040, this modification has possibility on either cysteine. All the LC-MS/MS raw data and searched results have been deposited to the ProteomeXchange Consortium via the PRIDE partner repository with the dataset identifier PXD042558 and 10.6019/PXD042558. We also revised the methods and results part in the revised MS accordingly.

a. If it is covalent, also demonstrate that preincubated UBA1-AF adduct retains the enhanced E2 interaction activity even when free AF is purified away. This is shown in the experiment in FigS1 for UFD but what about full length UBA1? Did the authors attempt this with full length UBA1?

Response: The suggested experiment has been performed and the new data is presented in Supplementary Fig. 1A, showing that preincubated and purified full-length UBA1-AF retains the enhanced E2 interaction activity.

b. Utilize LC/MS to demonstrate that other proteins in colPs or assay reactions are not being modified (E3, E2, etc).

Response: We appreciate the reviewer's suggestion and concern. However, our data do not support other proteins are modified by AF. Firstly, our SPR analysis

showed that AF does not bind to UBE2G2 (Fig. 1E), suggesting that AF does not directly modify E2 enzymes. Secondly, our in vitro ubiquitination assays demonstrated that AF enhances the activity of seven different E3 ligases from RING finger, HECT domain, RBX, and RBR families (Fig. 5A and S6A-D), indicating that AF does not specifically interact with a single E3 family or structure. Moreover, our data showed that AF enhances the interaction between UBA1 and UBA1's UFD and E2s in vitro in the absence of ubiquitin (Supplementary Fig. 1, 4), indicating that AF does not directly modify ubiquitin to enhance the binding. Therefore, based on our results, we believe that LC/MS analysis is not necessary to demonstrate that other proteins in the co-immunoprecipitation or assay reactions are not being modified by AF.

c. Request: the 5.19nM Kd (with 2 sig digits) is quite specific and is an important point. This reviewer is not an SPR expert, but is it possible to calculate a kD from a covalent binder at a level that is lower than the lowest measured concentration? What is the kD with the C1039A, E1037A and L954A?

Response: Thank the reviewer for the valid concern about KD calculation, which was also pointed out by Reviewer 1. Please see our response to reviewer 1.

Figure 2:

- Great to see the exploration and use of the point mutations to evaluate on mechanism/on target AF activity. These mutants are powerful tools, provided they do not affect UBA1 activity, and should be utilized throughout the experiments in other figures.

Response: Thank you for the suggestion. In response to the concern raised by reviewer #1 regarding the effect of UBA1 mutations on its activity and folding, we have observed that the L954A mutation reduces UBA1 activity in catalyzing Hrd1c-mediated ubiquitination in vitro, as depicted in the figure below.

Considering the location of the AF binding site at the UFD-E2 binding interface and its contacts with both UFD and UBE2G2, as predicted by our new model (Fig. 2D), mutations in the AF binding site may potentially affect AF binding, UBA1-E2 interaction, and UBA1 folding. These effects are difficult to be analyzed solely by biochemical and cell biology assays. Therefore, to fully elucidate the binding mechanism underlying AF-UBA1-E2 association, structural studies are ultimately

required, which we believe is beyond the scope of the present study. We have revised the manuscript accordingly.

To provide preliminary supports of our proposed model, we have now included only Figure 2F, which demonstrates that mutations of two key predicted binding residues, E1037 and E1049, diminish the effect of AF.

Based on the reviewer's comment and the aforementioned reasons, we have incorporated key mutations, specifically C1039A and C1040A, in other experiments as necessary (Fig. 2B, 3E, 7, and Supplementary Fig. 2, 6F). We have included new data showing that these mutants do not affect UBA1 activity, as indicated by ubiquitin charging to UBA1 in Supplementary Fig. 6F. Additionally, we have shown in Figure 3E that the C1039A mutant diminished AF-enhanced UBA1 interactions with two other E2s, UBE2A and UBE2L3. Furthermore, we have demonstrated in Figure 7B that AF has diminished activity in enhancing E3 activity in the presence of C1039A but not wild-type UBA1. Moreover, AF failed to enhance UBA1-UBE2L3 interaction and NHK ubiquitination and degradation in HCT116 cells expressing C1039A mutant created by genome editing of UBA1 gene using CRISPR/cas9 (Fig. 7).

- Fig 2E: blocking with IAA indicates it could be covalent but is not definitive proof that AF is modifying UBA1. Did the authors attempt LC/MS confirmation to ID the covalently modified peptide? This would be a critical result as mentioned above.

Response: Yes, please reference the detailed response above. We have performed HPLC-MS/MS experiments and confirmed that AF covalently modify either C1039 or C1040 in the UFD of UBA1 in the full-length UBA1. The result is presented in Supplementary Fig. 2. Our data support C1039 is the AF conjugation site since C1040 is buried in UBA1 structure and mutation of C1040 does not affect AF activity in enhancing UBA1-E2 binding and E3 activity.

Figure 3:

- This reviewer appreciates the intent of the coIP LC/MS study and appreciate the thoughtfulness of including the Sumo and UBA6/Ub pairings, but the statements require the C1039 pt mutants as important controls (if proven to be the covalent modification site). Showing that the C1039A point mutation fully abrogates the AF induced coIP phenomenon would greatly strengthen the claims of this manuscript.

Response: Thank you for the suggestion. We have now included new data in Fig. 3E showing that the C1039A mutation diminishes the AF-enhanced co-IP of UBA1 with two additional E2s, UBE2A and UBE2L3. Moreover, we have generated C1039A mutation in HCT116 cells by CRISPR/cas9 and demonstrated that the mutation abrogated AF activity in enhancing ERAD substrate NHK ubiquitination and degradation in cells and gp78-mediated ubiquitination in vitro (new Fig. 7), providing further evidence that C1039 is the target site of AF to enhance UBA1 interaction with E2s.

Figure 4:

- Making an argument about enhanced E2 trans-thiolation is tricky given how efficient these reactions often are. In the absence of AF, the E2 trans-thiolation shown appears very weak – this reaction is usually quite robust. Did the authors perform a time course or other kinetic experiments?

Response: We appreciate the reviewer's comment regarding the efficiency of E2 trans-thiolation reactions. We agree that these reactions are typically robust, and we acknowledge that the observed weak signal in the absence of AF is relative to that in AF-treated assays. As suggested, we have now performed a time course analysis of E2 charging, showing that AF promotes a time-dependent increase in ubiquitin charging to UBE2G2 (Supplementary Fig. 5). We hope that this additional data helps to clarify our findings.

- Was the experiment tested with pretreated and purified UBA1-AF to confirm any enhancement was through the mechanism the authors claim and not AF reacting with other proteins in the assay mixture?

Response: Thank you for your suggestion. We have addressed this concern by performing additional experiments using pretreated and purified UBA1-AF to confirm that the enhancement observed is through the mechanism we claim and not due to AF reacting with other proteins in the assay mixture. The new data is presented in Supplementary Fig. 1A, showing that preincubated full-length UBA1-AF retains the enhanced E2 interaction activity. Additionally, we used AF-pretreated and purified MBP-UFD in all the in vitro MBP-UFD pulldown assays in our first submission (Supplementary Fig. 1B & 4, and Fig. 2C), which was described the pulldown assay in detail in the Materials and Methods section in our first submission.

As to whether the enhancement was due to AF reacting with other proteins in the assay mixture, we believe that our data do not support this possibility. For example, the assay mixtures for 6His-UBA1 or MBP-UFD to pulldown UBE2G2 (Fig. 2C, Supplementary Fig. 1A), do not contain additional protein, whereas AF does not bind to UBE2G2 as demonstrated by SPR (Supplementary Fig. 3B). Thus, the sum of our data strongly indicates the binding site for AF is the UFD of UBA1, which has been confirmed by our mass spectrometry data (Supplementary Fig. 2).

Figure 5:

- Pleased to see the use of in vitro E3 auto ubiquitination assays.
- Missing control of compound binding refractory mutants in the assay.

Response: Thanks for raising this concern. We have performed this experiment and new data is now presented in Fig. 7B, showing that AF has diminished

activity in enhancing E3 activity in presence of the AF-binding refractory mutant C1039A.

- Please explain:

o why so little autoubiquitination of the E3s is shown in the absence of AF. Is this a blot exposure issue?

Response: Thank you for bringing up this issue. We apologize for any confusion caused by the low exposure of the blots. Yes, it is a blot exposure issue. Because the differences in ubiquitination activity with and without AF are so large for most of the E3s tested, we used low exposure to prevent too much overexposure of the AF plus reactions. We have now included high-exposure images below for the relevant E3s, which better illustrate the ubiquitination levels in the absence of AF for your review. Additionally, we have performed a new assay for RNF126 to show its activity in the absence of AF, and the new data replaced the old data in Fig. 5A. These results indicate the robustness of AF in increasing the activity of some of the E3s tested in vitro.

o Some E3s are low MW, others like E6AP are ~100kDa yet the poly Ub is mostly below the 95kDa marker suggesting free Ub chains or other proteins are being modified. Have the authors confirmed this is autoubiquitination of the actual E3?

- Recommend a time course be shown for at least 1 E2/E3 pair.

Response: Thank you for raising this concern. We apologize for any confusion caused by our inaccurate use of “autoubiquitination”. As the reviewer rightly pointed out, E3s can either autoubiquitinate or synthesize free ubiquitin chains or ubiquitinate substrate proteins. To address this issue, we have now included a Coomassie-stained gel image showing the sizes of the recombinant E3s or their fragments used in the in vitro assay (Supplementary Fig. 6A). The reviewer is right that the polyubiquitin chains in E6AP reaction are likely free ubiquitin chains or other proteins are being ubiquitinated. We have revised the text to refer to the assay as an in vitro ubiquitination assay, which measures E3 activity in general.

As the reviewer suggested, we have now included a time course study for gp78c and UBE2G2 in Supplementary Fig. 6D to demonstrate the dynamics of ubiquitination in this system.

Figure 6:

- AF is pleiotropic, need UBA1 compound binding point mutant controls to confirm that AF

effects on protein turnover are through the UBA1-AF/E2 mechanism being claimed:
o Recommended experiment: Engineer compound refractory pt mutations into UBA1 in cells (using CRISPR) and demonstrate that the effects of the compound on enhanced protein turnover are dependent on the UBA1 modification. As mentioned in the publication, AF is a known inducer of oxidative stress, ER stress and UPR – which could lead to effects on protein translation, chaperone induction, etc.

Response: We have performed this recommended experiment by generating AF refractory C1039A mutation in UBA1 in HCT116 cells. As predicted, AF is no longer able to enhance UBA1-E2 interaction and NHK ubiquitination and degradation in C1039A cells (Fig. 7A, C, D). Consistently, AF is also not able to enhance gp78-catalyzed ubiquitination in presence of C1039A mutant in vitro (Fig. 7B).

- Did the authors perform a covalent modification scan (e.g. like what D. Nomura or B. Cravatt do) to see which proteins are modified in the cell? What if deubiquitylating enzymes such as the thiol dependent ubiquitin specific proteases are being modified... this could enhance protein turnover too through an unrelated mechanism.

Response: We did not perform a covalent modification scan in this study. However, a similar study has been reported in which a combination of Thermal-range Thermal Proteome Profiling (TR-TPP), Functional Identification of Target by Expression Proteomics (FITeXP) and multiplexed redox proteomics was used for deconvolution of auranofin targets (Saei A et al. Redox Biol. 2020 May; 32: 101491). The cumulative sum of individual target rankings in four different types of analysis (FITeXP, TR-TPP in cells and lysate, as well as deep redox proteomics) identified four targets, including Thioredoxin reductase 1, a known AF target. No DUB protein was identified as AF target. Although UBA1 is not among the four targets, the study indeed shows that UBA1 is one of the top proteins stabilized by AF treatment in Thermal-range Thermal Proteome Profiling (TR-TPP) assay. This study supports our conclusion that UBA1 is a target for AF.

General comment:

The rate limiting step of protein ubiquitylation is at the level of the substrate/E3 protein, not typically at the level of the E1 activating enzyme. This manuscript would infer that E2-Ub thioester levels are limiting, which is to this reviewer's knowledge has not been shown to be the case.

Response: Thank you for your comment. We agree that it is generally accepted that the rate-limiting step in protein ubiquitination occurs at the level of the substrate/E3 protein, and the E1 activating enzyme is not typically considered as a rate-limiting step. However, there is evidence suggesting that E1 activity can also play a key role in regulating ubiquitination activity. For instance, it has been reported that E1 activity can be rate-limiting in ubiquitination in certain cell types, such as lens cells (Shang F. et al. 1997, 272:23086). In addition, E3 has to bind to

both substrate and ubiquitin-loaded E2 to catalyze substrate ubiquitination. Ubiquitin-loaded E2 is known to have higher affinity to E3 than ubiquitin-free E2 (Metzger MB, et al. Biochim Biophys Acta. 2014, 1843: 47). Moreover, UBA1 provides activated ubiquitin for hundreds of E3s to catalyze ubiquitination of thousands of substrate proteins. Therefore, AF enhances UBA1 activity to generate more ubiquitin-loaded E2s, which will promote assembly of E3, ubiquitin-loaded E2, and substrate complex, thereby facilitating E3 to catalyze substrate ubiquitination as we have demonstrated for seven different E3s in vitro and four different proteasomal substrates in cells.

Moreover, reduced UBA1 activity due to somatic mutations of UBA1 gene has been shown to decrease ubiquitination in hematopoietic stem cells and cause VEXAS syndrome (Beck D. et al. NEJM, 2020, 383: 2628). Decreased UBA1 activity due to germline mutations in UBA1 gene causes X-linked spinal muscular atrophy (Dlamini N., et al. 2013, 23: 391). Therefore, it is evident that variations in UBA1 activity is critical for ubiquitination and the maintenance of cellular homeostasis. Our findings demonstrate that compounds like AF can enhance ubiquitin loading to E2s and promote ubiquitination, which may have potential therapeutic applications in diseases related to aberrant ubiquitination.

We appreciate your comment and hope that our response adequately addresses your concerns.

Minor points:

- Should always include the chemical structure in a manuscript.

Response: AF structure is now shown in Fig. 1A.

- First sentence of abstract: "UBA1 is the primary E1 ubiquitin-activating enzyme, regulating stability and function of numerous proteins via initiating their ubiquitination". Recommend rewording. The ubiquitin proteasome system (UPS) not UBA1 per se regulates the stability and functions of cellular proteins. UBA1 ensures that sufficient activated ubiquitin is available for the UPS to perform its cellular function.

Response: Thanks for the comment. The sentence has been re-worded as "UBA1 is the primary E1 ubiquitin-activating enzyme responsible for generation of activated ubiquitin required for ubiquitination, a process that regulates stability and function of numerous proteins."

Reviewer #3 (Remarks to the Author):

Small molecules targeting ubiquitin-mediated proteasomal degradation pathway remain poorly defined. In this manuscript, by using in vitro assays, Yan et al verify auranofin (AF), a drug approved for the treatment of rheumatoid arthritis, is an enhancer specifically for the E1 ubiquitin-activating enzyme UBA1 and facilitates ERAD as well as outer mitochondrial

membrane-associated degradation. The study is overall very interesting and solid, and is well suited to be published in NC.

However, some conclusions are not clear and robust, and some additional key experiments may be needed to make the study more comprehensive and impactful.

We are grateful to the reviewer for expressing support for publication of our manuscript in Nature Communications, and for providing valuable suggestions to enhance its quality.

Major points:

The major conclusion of this paper is that AF promotes protein ubiquitination and degradation in cells, including the ERAD process. The authors demonstrated this using the model ERAD substrates NHK-HA and CD3d-HA (Fig. 5D-5H). However, the data is not very convincing with several concerns are: First, UBA1 is not known to be involved in ERAD. Although the Fig 5C showed that UBA1 can enhance the autoubiquitination of ERAD E3 gp78, it is not sufficient to conclude that E3 activity is enhanced. It is critical to show that UBA1 is required for ERAD by generating the UBA1 KO cells to examine whether known ERAD substrates (OS9, IRE1a and CD147) are stabilized and accumulate. Moreover, HRD1 is a much better known E3 ligase involved in ERAD, hence autoubiquitination assay should also be performed for HRD1, not just gp78.

Response: Thank you for these constructive comments. We have addressed your concerns by adding a set of new experiments.

1) We investigated the role of UBA1 in ERAD and added new data showing that inhibition of UBA1 with the UBA1 inhibitor TAK-243 inhibited dislocation of the luminal substrate NHK and membrane spanning substrate CD3delta (Supplementary Fig. 7), which is consistent with our other results indicating that UBA1 is involved in ERAD.

2) We attempted to generate UBA1 KO cells using CRISPR/Cas9 in both 293T and HCT116 cells. We were unable to recover any clone with KO of UBA1. It is highly likely that UBA1 is an essential gene for cell survival as deletion of the UBA1 gene has been shown to be lethal (Kulkarni M and Smith HE, PLoS Genet. 2008; 4: e1000131; McGrath JP et al. EMBO J. 1991; 10: 227), and is classified as a 'common essential' gene in the cancer dependency map in which 1095 out of 1095 cell lines tested were not tolerant of knockout (<https://depmap.org/portal/gene/UBA1?tab=overview>).

3) In vitro ubiquitination assay has now been performed for Hrd1, and the new data on the effects of AF on Hrd1c activity is now presented in Supplementary Fig. 6B, showing that AF potently enhances Hrd1c E3 activity in vitro as demonstrated for other E3s.

Some experiments require important controls: The mRNA levels should be shown in Fig 5D and 5H.

Response: New data on mRNA levels is now presented in Supplementary Fig. 8D, E, showing no changes in mRNA levels.

Fig 6A should test NHK-HA and CD3d-HA.

Response: As also pointing out by reviewer 2, the non-specific band in Fig. 6A is confusing. We have now included new data in Fig. 6A, showing that AF increases endogenous Mcl1 degradation.

Fig 5A should include one negative control, for example, changing E1 (UBA1) to UBA6 which is not affected by AF, and should show total protein levels of E1, E2 and E3.

Response: Thank you for your comment. The control experiment with UBA6 is now shown in Supplementary Fig 6E. We have now also included a figure (Supplementary Fig. 6A) showing the recombinant E3s produced in-house used in our assays. UBA1 and E2 blots were also added to Fig. 5A. Additionally, we have included the product figure below cited from Bio-technne from where we purchased the RBX E3 complex (https://www.bio-techne.com/p/proteins-enzymes/recombinant-human-elongin-b-elongin-c-vhl-cul2-rbx1-cf_e3-655) for you to review. We appreciate your suggestion and hope that these additions will improve the clarity and completeness of our study.

RBX E3 complex product figure cited from Bio-technne (https://www.bio-techne.com/p/proteins-enzymes/recombinant-human-elongin-b-elongin-c-vhl-cul2-rbx1-cf_e3-655).

Fig 1C-D is quite confusing and more explanation and description are needed - What do these lines mean and why do two lines look comparable in Fig 1C?

Response: We apologize for the confusion that Fig. 1C caused. The black line is the signal line that shows how the SPR signal changes over time as AF interacts with UBA1 on the sensor surface. The red line is fit line, also known as the kinetic fit or sensorgram fit, that is a curve that is fitted to the experimental data points of the signal line using a mathematical model. It helps to extract quantitative information about the binding kinetics and affinity of the interaction between AF and UBA1. A brief description of this information is now added in the legend for Fig. 1C-D (now Supplementary Fig. 3 in this revised manuscript).

For Fig 1E, UBA1 KO 293T should be included as a control to make the conclusion more solid.

Response: Thank you for your comment. We attempted to generate UBA1 KO cells using CRISPR/Cas9 in both 293T and HCT116 cells. We were unable to recover any clone with KO of UBA1. In contrast, we were able to generate UBA1(C1039A) mutant HCT116 cell lines by CRISPR/cas9 (Fig. 7). It is highly possible that UBA1 is an essential gene for cell survival as deletion of the UBA1 gene has been shown to be lethal (Kulkarni M and Smith HE, PLoS Genet. 2008; 4: e1000131; McGrath JP et al. EMBO J. 1991; 10: 227) and the Cancer Dependency Map (depmap.org). Moreover, UBA1 gene is classified as a ‘common essential’ gene in the cancer dependency map in which 1095 out of 1095 cell lines tested were not tolerant of knockout (<https://depmap.org/portal/gene/UBA1?tab=overview>).

For Fig 3A, please explain for the first two panels.

Response: Thank you for pointing out this confusion. The first two panels show $-\log_{10}(P\text{-Value})$ and fold change (FC), respectively, in UBA1-E2 binding between control and AF-treated cells. The legend has been updated accordingly to clarify.

For Fig 3C, it would be better to confirm the interactions using endogenous IP of anti-UBA1 antibody as shown in Fig. 1A.

Response: Thank you for your comment. We have added new data showing that AF enhances endogenous UBA1 interactions with two more E2s in Fig. 3D. We hope this new data has addressed your concerns.

Minor Points:

1) The rationale for testing AF is missing.

Response: In recent years, we have been focusing on discovery of small molecule modulators of ERAD. The searching process is lengthy and complex, particularly for AF, and therefore, we choose to simplify the rationale.

1) Show the domain structure for UBA1 in Fig 2.

Response: UBA1 domain structure is now shown in Fig. 2D.

2) Which cell type was used in Fig 2A? The cell type of 293T cells was shown in the text but Hela cell was shown in the figure legend.

3) Which cell type was used in Fig 2D? No mention in both text and figure legend.

Response: Thank you for bringing up these mistakes. The cell type used in Fig. 2A and 2D (now Fig. 2B and 2E in this revised manuscript) was 293T cells. have corrected this error.

4) Statistic analysis for Fig 5G.

Response: Statistic analysis has been added for Fig 5G.

5) What is the E1 in Fig 5A? Please mention it in the figure and the figure legend.

Response: E1 used in Fig 5A is UBA1 and the figure label has been revised to clarify.

REVIEWERS' COMMENTS

Reviewer #1 (Remarks to the Author):

The authors have fully addressed the concerns I raised in the previous round and, as far as I can tell, have also addressed the concerns of reviewers 2 and 3. The addition of the MS data, improved docking results with corroborating site directed mutagenesis, relegating the SPR data to the supplement, new and or improved biochemical and cell biological data have, in my opinion, considerably strengthened the manuscript. Consequently, I recommend publication of the manuscript in Nature Communications.

Reviewer #2 (Remarks to the Author):

I would like to thank the authors for their responses to the reviewers comments; and for revisions and the generation of new data to strengthen the publication.

Reviewer #3 (Remarks to the Author):

The authors have nicely addressed all my previous concerns. Fig S6B should be in the main figure. i have no others issues.

We are grateful to the reviewers for their constructive comments, which has greatly improved our work.

REVIEWERS' COMMENTS

Reviewer #1 (Remarks to the Author):

The authors have fully addressed the concerns I raised in the previous round and, as far as I can tell, have also addressed the concerns of reviewers 2 and 3. The addition of the MS data, improved docking results with corroborating site directed mutagenesis, relegating the SPR data to the supplement, new and or improved biochemical and cell biological data have, in my opinion, considerably strengthened the manuscript. Consequently, I recommend publication of the manuscript in Nature Communications.

Response: We would like to thank the reviewer for assessing our manuscript and for recommendation for its publication in Nature Communications.

Reviewer #2 (Remarks to the Author):

I would like to thank the authors for their responses to the reviewers comments; and for revisions and the generation of new data to strengthen the publication.

Response: We would like to thank the reviewer for reviewing our manuscript and providing detailed assessment of manuscript for improving our work.

Reviewer #3 (Remarks to the Author):

The authors have nicely addressed all my previous concerns. Fig S6B should be in the main figure. i have no others issues.

Response: We would like to express our gratitude for reviewing our manuscript. We appreciate your feedback and would like to address the concern regarding Fig. S6B not being included in the main figure.

The reason we did not include Fig. S6B in the main figure was primarily due to space limitations in the main figure. Since the main figure has representatives of all major classes of E3 ubiquitin ligases, we believe that the suggested re-arrangement will not affect the quality and conclusion of the figure, which is that auranofin enhanced the activity of all major classes of E3s in vitro.

Once again, we thank you for your valuable input and consideration.